

# The Plio-Pleistocene climatic evolution as a consequence of orbital forcing on the carbon cycle

Didier Paillard[1]

[1]LSCE, Centre d'Etudes de Saclay, 91191, Gif-sur-Yvette, France

*Correspondence to*: Didier Paillard (didier.paillard@lsce.ipsl.fr)

**Abstract.** Since the discovery of ice ages in the XIX[th] century, a central question of climate science has been to understand the respective role of the astronomical forcing and of greenhouse gases, in particular changes in the atmospheric concentration of carbon dioxide. Glacial-interglacial cycles have been shown to be paced by the astronomy with a dominant periodicity of 100 ka over the last million years, and a periodicity of 41 ka between roughly 1 and 3 million years before

present (MyrBP). But the role and dynamics of the carbon cycle over the last 4 million years remain poorly understood. In particular, the transition into the Pleistocene about 2.8 MyrBP or the transition towards larger glaciations about 0.8 MyrBP (sometimes refered as the mid-pleistocene transition, or MPT) are not easily explained as direct consequences of the astronomical forcing. Some recent atmospheric $CO_2$ reconstructions suggest slightly higher $pCO_2$ levels before 1 MyrBP and a slow decrease over the last few million years (Bartoli et al., 2011; Seki et al., 2010). But the dynamics and the climatic role

of the carbon cycle during the Plio-Pleistocene period remain unclear. Interestingly, the $d^{13}C$ marine records provide some critical information on the evolution of sources and sinks of carbon. In particular, a clear 400-kyr oscillation has been found at many different time periods and appears to be a robust feature of the carbon cycle throughout at least the last 100 Myr (eg. Paillard and Donnadieu, 2014). This oscillation is also visible over the last 4 Myr but its relationship with the eccentricity appears less obvious, with the occurrence of longer cycles at the end of the record, and a periodicity which therefore appears

shifted towards 500-kyr (cf. Wang et al., 2004). In the following we present a simple dynamical model that provides an explanation for these carbon cycle variations, and how they relate to the climatic evolution over the last 4 Myr. It also gives an explanation for the lowest $pCO_2$ values observed in the Antarctic ice core around 600-700 kyrBP. More generally, the model predicts a two-step decrease in $pCO_2$ levels associated with the 2.4 Myr modulation of the eccentricity forcing. These two steps occur respectively at the Plio-Pleistocene transition and at the MPT, which strongly suggests that these transitions

are astronomicaly forced through the dynamics of the carbon cycle.

## 1 Introduction

The idea that the orbital parameters of the Earth may influence climate has a long history, linked mostly to the development of theories of ice ages (eg. Paillard, 2015). But it is quite clear from geological records, that astronomical climatic variations are occuring throughout the Earth history, with or without ice being present on Earth. It is therefore certain that the





astronomical parameters are influencing climate through other mechanisms than the growth and decay of ice sheets. This is for instance well-known concerning records of monsoons or records of low latitude precipitations, which are strongly influenced by precession. A very illustrative example is given by the Mediterranean sapropels (Lourens et al., 1996; Hilgen et al., 1999) which are used to calibrate the $^{40}Ar/^{39}Ar$ radiochronometers (Kuiper et al, 2008). Similarly, a 400-kyr oscillation

is observed in the $d^{13}C$ of the foraminifera recovered from marine records, throughout most of the Cenozoic (Pälike et al., 2006; Cramer et al., 2003; Sexton et al., 2011; Billups et al., 2004; Wang et al., 2010). This oscillation is present in the benthic records, but also in many planktic ones, suggesting that this $d^{13}C$ variations are linked to global ocean $d^{13}C$ changes. This persistent oscillation was recently used to reconstruct the evolution of the Earth's carbon over the last 100-Myr (Paillard and Donnadieu, 2014). A key difficulty is to understand the dynamics of this cycle. In particular during the last million

years, these oscillations appear to stretch and the relationship with eccentricity becomes less clear (eg. Wang et al., 2004; 2010), as illustrated on Figure 1.

Before 1 MyrBP when ice sheets remained medium sized, the cyclicity appears locked to eccentricity, with high eccentricity values associated with decreasing or low values in $d^{13}C$. This phase relationship appears consistent with earlier time periods, with the chronology of Cenozoïc marine cores being sometimes based on the association of high eccentricity and low $d^{13}C$

values (eg. Paelike et al., 2006; Cramer et al., 2003). A simple deduction is that, most probably, the dynamics behind this oscillation is essentially stable and linked to the astronomical forcing before 1 MyrBP, but it is strongly disturbed by the large Quaternary glaciations afterwards. This observation has major implications on the possible mechanisms, as we will see further on.

There is no consensus on the cause of these $d^{13}C$ oscillations, but monsoons or the associated low latitude precipitations are

known to respond to precessional forcing, and therefore to be modulated by the 400-ky eccentricity cycles. Still many factors may contribute to the evolution of the carbon cycle on these time scales, like erosion, vegetation dynamics, ocean biogeochemical or dynamical changes. It was therefore suggested that the $d^{13}C$ cycles could be caused by the modulation of weathering in monsoonal regions (Paelike et al 2006) or by ecological shifts in marine organisms, possibly linked to nutrient availability (Wang et al, 2004; Rickaby et al, 2007). It is worth emphasizing that during the last million years, if the link with

eccentricity is less obvious, there are clear indications that these $d^{13}C$ shifts are associated to major changes in the Earth carbon cycle. For instance, carbonate deposition exhibits major changes, well correlated with these global $d^{13}C$ changes (Bassinot et al, 1994 ; Wang et al, 2004), and the record of atmospheric $pCO_2$ from Antarctic ice cores also shows a 10 to 20 ppm long term modulation with a minimum level around 0.6-0.7 MyrBP and a maximum around 0.3-0.4 MyrBP (Lüthi et al, 2008) in phase with the long term carbonate preservation cycle. A mechanistic modeling of these 400-ky to 500-ky cycles is

therefore a critical missing element in our understanding of climate-carbon evolution over the Plio-Pleistocene period.

With a simple ocean box model (Russon et al, 2010), it was shown that silicate weathering alone could not account for the simultaneously observed rather large $d^{13}C$ changes (> 0.4‰) and rather small $pCO_2$ variations (< 20 ppm) in this frequency band during the last million years. Furthermore, with silicate weathering only, the model-predicted phase relationships were also inconsistent with observations of $d^{13}C$, carbonate deposition and $pCO_2$. Changes in organic matter fluxes are therefore a





necessary ingredient in order to account for the observed rather large $d^{13}C$ changes. A possible mechanism could therefore be linked to ocean organic matter burial, associated to changes in nutrient supply or ecological shifts (Rickaby et al, 2007). But it is then very difficult to explain why this mechanism would change drastically with the occurrence of major glaciations, as suggested on Fig. 1. We will therefore build our model on a different perspective, involving a more direct link between

monsoons and organic matter burial, that should be strongly affected by sea level changes.

Organic matter burial takes place mostly on the continental shelves. Recent re-assessments of riverine carbon fluxes to the ocean have emphasized the role of the erosion of continental organic carbon in the overall balance (eg. Galy et al., 2007; Hilton et al., 2015). When investigating the influence of monsoons on the carbon cycle, it is natural to have a closer look at river discharges in monsoonal areas. Carbon budgets on major present-day erosional systems have provided some contrasted

results, with riverine organic matter being either a net carbon source for the ocean (Burdige, 2005), or a net sink through organic carbon burial in sedimentary fans (Galy et al., 2007). The first study was mostly based on the Amazon basin, while the second estimation is from the Himalayan system. The differences are likely linked to different river basin configurations and different sedimentary deposition dynamics. This dramatically highlights the impact of geomorphology on terrestrial organic carbon burial, and suggests that the long term global balance might be different in a context of large glacial-

interglacial sea level variations like the last million years, when compared to earlier periods with much smaller sea level changes. Our conceptual model is therefore built on the impact of monsoon-driven terrestrial organic matter burial on the global carbon cycle.

## 2 Conceptual Model

We are interested by the evolution of the global Earth carbon, that is the carbon content of the atmosphere, the ocean and the

biosphere, which amounts today approximately to $C = 40,000$ PgC (petagrams of carbon, ie. $10^{15}$gC). This evolution results from possible imbalances between the volcanic inputs V, the oceanic carbonate deposition flux D associated to silicate weathering and its alkalinity flux to the ocean W, and the organic carbon burial B. Our model equations are:

(1a)    $dC/dt = V - B - D$

(1b)    $dA/dt = W - 2D$

where the second equation represents the alkalinity balance. On time scales larger than several millennia, carbonate compensation will restore the oceanic carbonate content. Therefore, to first order, we can write :

$$d[CO_3^{2-}]/dt = d(A-C)/dt = 0 = W - D - V + B.$$

Solving for D, this leads to the long-term evolution equations for carbon:

(2a)    $dC/dt = 2(V - B) - W$

(2b)    $d\delta^{13}/dt = (V(-5-\delta^{13}) - B(-25-\delta^{13}))/C$





where the second equation corresponds to the $^{13}C$ budget, with $\delta^{13}$ the isotopic composition $\delta^{13}C$ of marine carbonates, and where we are assuming a constant -5‰ volcanic source, as well as a constant -25‰ organic matter fractionation. We also assume that carbonate compensation has no significant impact on ocean $\delta^{13}C$.

For simplicity, we will assume that the main stabilizer of the carbon system is the silicate weathering, with a fixed relaxation time $\tau_C$, ie. $W = C/\tau_C$. Solving the present-day equilibrium with $\delta^{13}_{Eq} = 0‰$ as a typical value for carbonates, we easily deduce typical equilibrium values for the fluxes : $B_0 = V/5$ ; $C_{Eq} = (8/5)\,\tau_C\,V = 40,000$ PgC. If we assume a relaxation time $\tau_C$ of 200 kyr (Archer et al., 1997), we obtain $V = (5/8)\,C_{Eq}/\tau_C = 125$ TgC/yr and $B_0 = 25$ TgC/yr. For a larger value $\tau_C = 400$ kyr (Archer et al. 2005), we would get $V = 62$ TgC/yr. There is no consensus on the actual total carbon emissions from volcanism (including all aerial and submarine sources), but these values for V (or $\tau_C$) span more or less the range of current

estimates from about 40 to 175 TgC/yr (Burton et al, 2013).

In order to translate the total carbon content C into an equivalent $pCO_2$ level, we will use the simple scaling: $pCO_2 = 280\,(C/40,000)^2$ (in ppm). To reproduce a multi-million year trend, we need to add one explicitly in the weathering relaxation: $W = C/\tau_C = (\Delta C + C_{Eq} - \gamma\,t)/\tau_C$ , with the coefficient $\gamma$ chosen to obtain the desired $pCO_2$ levels at the start of the simulation, ie. about 350 ppm at 4 MyrBP, according to current estimates (Bartoli et al., 2011; Seki et al., 2010).

In the following, we are describing how carbon burial B should vary with monsoons, and what consequences these variations have on the total carbon content C as well as on carbonate isotopes $\delta^{13}C$. In order to represent the monsoon's response to astronomical forcing, we introduce a simple truncation of the precessional forcing:

$$F_0(t) = \max(\,0,\, - e \sin\omega\,)$$

where $e$ is the eccentricity and $\omega$ the climatic precession.

Indeed, soil erosion or sediment transport are dominated by intense events, not by the average climate. Such a non-linear response can be mimicked in a simple way by the above expression that accounts only for positive monsoonal forcing, not for negative one. Consequently, the model will be influenced by the amplitude modulation of the precessional forcing, ie. the eccentricity. To avoid useless parameters, we furthermore introduce the normalization:

$$F = F_0/\mathrm{Max}(F_0) - <F_0/\mathrm{Max}(F_0)>$$

which results in a precessional forcing F(t) with amplitude one and zero mean.

We implicitly account for a slow terrestrial organic carbon reservoir (soil) as "buried organic carbon". It is reasonable to assume that monsoon, or enhanced precipitation will favor primary production and soil formation. But this recent soil and also possibly older soils will be eroded and transported to the ocean through enhanced river discharges. If the corresponding carbon is remineralized in the ocean without too much burial in the alluvial fan, the net perturbation of the burial flux is

likely to be negative (ie. net "old" soil erosion and remineralization). We will refer to this case as the "Amazon-like" situation, with the perturbation F(t) being substracted to the baseline burial $B_0$ ($B = B_0 - a\,F(t)$). In contrast, if most of the organic carbon is buried and preserved in the sediment, then the perturbation is likely to be positive, since it induces a net "recent" soil formation and burial. We call this the "Himalayan-like" situation, with now $B = B_0 + a\,F(t)$. Before 1 MyrBP





and the associated major sea level changes, the river fans and continental shelves should evolve mostly in a progradational way (see scheme on Fig. 3), a situation which *a priori* favors organic carbon remineralization, while aggradational situations are likely to be more frequent in the late Pleistocene, with therefore a possible temporary reversal of the organic carbon burial.

## 3 Results

Our first simulations, with $B = B_0 - a\,F(t)$, correspond to a perpetual "Amazon-like" situation. They correspond to experiment *a* (black lines) with no trend in the total carbon, and experiment *b* (blue lines), with an explicit linear trend in carbon. As can be seen on Fig.2, we obtain a surprisingly good match between the simulated and observed $\delta^{13}C$, with very similar cycles (the $\delta^{13}C$ black and blue curves are superimposed and almost undistinguishable). The two main exceptions

occur at about 0.3 and 2.3 MyrBP, with the simulated $\delta^{13}C$ being significantly too high. In experiment *a* (black lines), $pCO_2$ is oscillating around its equilibrium value, with two significant negative excursions occurring near 2.5 MyrBP and near 0.5 MyrBP. These lower values are directly linked to the ~2.4 Myr modulation of eccentricity. Obviously, with fixed or periodic parameters, this model cannot simulate a long term decreasing trend in carbon. When explicitly adding such a linear decreasing trend (experiment *b*, blue lines), the two minima described above become two decreasing steps. The first one,

occurring around 2.8 MyrBP, is coincident with the Plio-Pleistocene transition and the development of Northern hemisphere glaciations. The second one near 0.8 MyrBP is coincident with the Mid-Pleistocene transition (MPT) and the significant amplification of glaciations. Note that the timing of these two steps is directly linked to the astronomical forcing: it does not depend at all on the specifics of the trend that we used here. Two similar $pCO_2$ decreasing episodes are also seen in the data (Figure 1) though it is difficult to associate them with a precise timing, due to the difficulties to reconstruct accurately $pCO_2$

from indirect proxies.

In order to account for the observed departure of the $\delta^{13}C$ oscillations from a simple eccentricity forcing, we need to introduce a retroaction of Quaternary sea level changes onto the sedimentary dynamics of alluvial fans and continental shelves, and consequently onto organic carbon burial. As explained above, we will reverse the sign of our burial flux perturbation, and change it into $B = B_0 + a\,F(t)$ when some conditions are met on the geomorphology of river outputs. In

particular it is necessary to account for a changing reservoir size that can be filled with sediments in an aggradational way. Indeed, at the first major sea level drop, rivers are incising though the river and fan bedrock, thus providing room for the accumulation of sediments loaded with organic carbon. This volume should be filled progressively with sedimentary organic carbon up to a point when further river incision, and consequent aggradation of sediment, do not affect the global organic carbon anymore but only move sedimentary carbon from one place to the other. In other words, we will assume that the

global "Himalayan-like" situation (ie. net organic carbon burial) is only a transient situation, linked to the first occurrence of a sea level minima. In order to illustrate this mechanism, we add a new equation for the slow geomorphological reservoir S for organic carbon in river beds or river fans. We define its maximal size $S_{MAX}$ from the observed sea level changes (Lisiecki



and Raymo, 2005) by finding the previous sea level minima $z_{MIN}$ (ie. the lower envelope) with the scaling $S_{MAX} \sim z_{MIN}^3$ since it represents a volume of sediment (see Fig. 2).

$$\text{(3a)} \quad \text{if } S < S_{MAX}: \qquad dS/dt = b\, F_0(t)$$
$$\text{otherwise:} \qquad S = S_{MAX}$$

In other words, the sedimentary organic carbon reservoir S grows at the pace of the above mentionned astronomical perturbation $F_0(t)$ up to its maximal size $S_{MAX}$. When S is small compared to the maximal reservoir size $S_{MAX}$, then the aggradational scheme is favoured, with river beds and deltaïc net organic carbon burial. But when S is close to its maximum value, we switch back to a mostly progradational sedimentation scheme, meaning that potential sea level changes will no more affect net global organic carbon burial :

$$\text{(3b)} \quad \text{if } S < 0.85\, S_{MAX}: \qquad B = B_0 + a\, F(t)$$
$$\text{otherwise:} \qquad B = B_0 - a\, F(t)$$

Using this simple crude criteria, we obtain the results show on Fig. 2 (experiment $c$, red lines). As expected, this simple model does switch from the background "Amazon-like" or progradational burial mode to a "Himalayan-like" or aggradational mode, after each significant sea level drop, and most notably at two time periods, the first one between 2.4 and

2.5 MyrBP (as a consequence of the Plio-Pleistocene transition) and the second and largest one between 350 and 650 kyrBP (as a consequence of the MPT). This allows for a much better agreement with measured $\delta^{13}C$ around 0.3 and 2.3 MyrBP, while the first simulations were systematically too high at this time. We also simulate correctly the $\delta^{13}C$ maximum around 500 kyrBP and the occurence of two broad "500 kyr" cycles over the last million years. With this burial mode switching mechanism, we are also able to predict an absolute minimum in carbon content, or long-term $pCO_2$, around 600 kyrBP, in

rather good agreement with the long term trend of $pCO_2$ measured in Antarctic ice cores. Indeed, $pCO_2$ from the Dome C record is about 5 to 10 ppm lower before the MPT (between 400 and 800 kyrBP), which is also what we obtain in our experiment $c$.

In order to further improve the qualitative match between measured and simulated $\delta^{13}C$, we can also add artificially a component linked to glacial-interglacial cycles. This is done by adding the detrended sea level LR04 curve to $\delta^{13}C$ obtained

from experiment $c$ (orange line on figure 2). This allows to account for the significant 100-kyr periodicity seen in the carbon isotopic record, and usually attributed to glacial-interglacial changes in the global biospheric size (eg. Shackleton, 1977).

## 3 Discussion

In order to reproduce the observed amplitude of the 400-kyr oscillations in the $\delta^{13}C$ records (about 0.5‰), the strength of the forcing $a$ needs to be of the same order than the baseline value $B_0$. This is a robust feature, which does not depend on model

setting or parameters. The observed 400-kyr signal in $\delta^{13}C$ records therefore requires major changes in the organic carbon burial, with almost no global net burial, or even net oxidation episodes, during maxima of precessional forcing. This strong forcing therefore implies significant oscillations in the Earth carbon cycle for this time frequency, up to 4 or 5% in total



carbon content. This is translated here into 10 to 20 ppm variations of $pCO_2$ using our simple scaling, but it is very likely that these changes would be much larger, when accounting for interactions between $pCO_2$ and climate. Indeed, colder climates are more favorable to oceanic carbon storage, as observed during the last glacial cycles.

It was already noted (Wang et al., 2004) that the climatic evolution since the last million years, in particular the Mid-Pleistocene Transition (MPT, about 0.8 MyrBP) and the Mid-Brunhes Event (MBE, about 0.4 MyrBP) were associated with the carbon isotopic maxima referred as $\delta^{13}Cmax$-II and $\delta^{13}Cmax$-III. This was a strong indication of a possible causal link between the long-term well-recognized eccentricity forcing on the carbon cycle and the Plio-Pleistocene climatic evolution. There is therefore a strong incentive to build a mechanistic astronomical theory of the carbon cycle. But a prerequisite towards understanding this long-term precessionally forced carbon cycle and its climatic consequences is to explain the observed changes during the Quaternary, in terms of $\delta^{13}C$ and simultaneously in the atmospheric $CO_2$ levels (Lüthi et al, 2008). The model results outlined above are a first step in this direction.

As detailed above, the fact that the 400-kyr carbon isotope cycle is perturbed during the Pleistocene strongly points towards a major role for organic matter burial over continental shelf areas being affected by sea-level changes. Obviously, this model is far too simple to represent faithfully the complexities of sedimentary dynamics in coastal areas, its consequences on organic matter preservation, on carbon cycle and ultimately on climate. Besides, we provide here no explanation for the prescribed multi-million year decreasing trend in $pCO_2$. There is unfortunately no clear consensus on the actual mechanisms involved, though this trend has been often attributed to long-term changes in continental weathering linked either to mountain uplift (Raymo and Ruddiman, 1992), to continental drift or mantle degassing rate (Lefebvre et al., 2013). Furthermore, we considered only sea-level changes as a potential feedback on organic matter burial in coastal areas. Obviously, many other important climatic feedbacks would also play a role: for instance increased temperatures would probably reduce net primary production as a consequence of increased stratification, and therefore reduce organic carbon deposition in coastal sediments, but it would also decrease oxygen concentrations and consequently would favor organic matter preservation. Similarly, stronger monsoons events would enhance the delivery of nutrients to the continental shelves, and therefore biological productivity. This would in addition deliver more fine-grained clay minerals that are necessary to seal and preserve organic matter from oxidation. This would work opposite to our continental soil-carbon mechanism for which enhanced monsoons lead to more organic carbon oxidation in agreement with the isotopic records. But, as a proof of concept, our model is chosen as minimalistic as possible. It does not attempt to include all potentially important mechanisms.

## 4 Conclusion

Our basic assumptions are primarily based on recent re-assessments of riverine organic carbon inputs to the ocean. With the above conceptual model, we demonstrate that simple mechanistic assumptions can account for the major patterns of the observed global evolution of carbon and carbon isotopes over this time period: First, enhanced precessional forcing linked to high eccentricity leads to more continental organic carbon been washed out and remineralized, therefore a net decrease in overall organic carbon burial. Second, this mechanism is temporarily reversed following major sea-level drops associated



with glaciations. This model accounts for the persistent 400-kyr oscillation observed in $^{13}$C records during the Cenozoïc, but also for its change during the last million years. It also suggests the occurrence of possibly significant $CO_2$ drops at about 0.8 MyrBP (Mid-Pleistocene transition) and at about 2.8 MyrBP (Plio-Pleistocene transition), that would ultimately link the timing of these transitions to the astronomical forcing. Our model also provides a possible explanation for the puzzling shifted level in the $CO_2$ records associated with the MBE

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





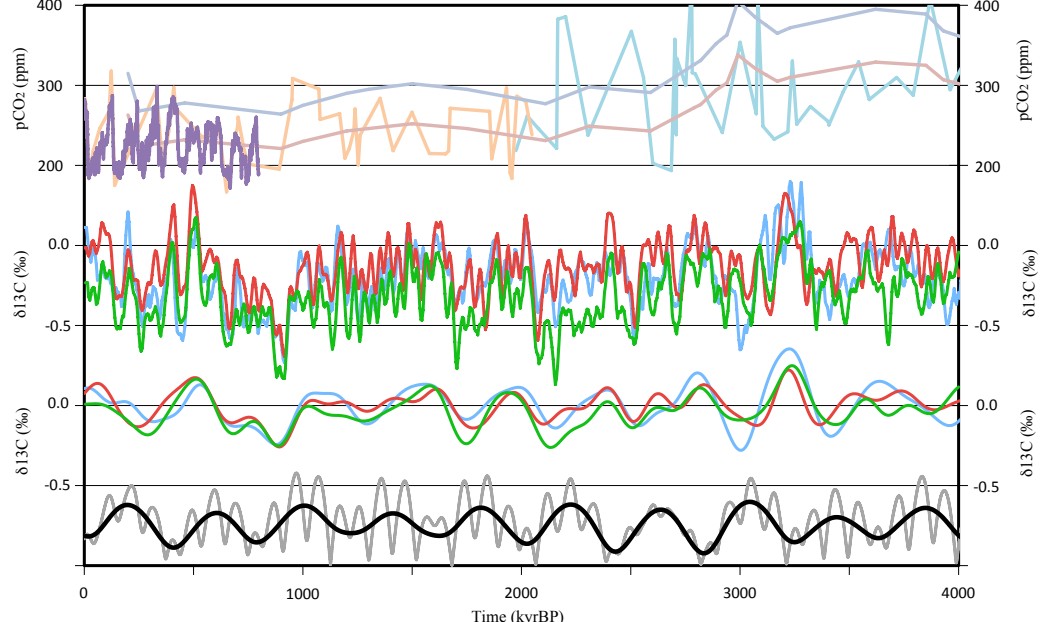

**Figure 1: From top to bottom: — pCO$_2$ records from Antarctic ice cores (purple: Lüthi et al, 2008); from boron isotopes in marine cores (orange: Hönisch et al, 2009; light blue: Bartoli et al., 2011) and alkenone isotopes (pink and blue lines for the min and max envelope, from Seki et al, 2010). — δ$^{13}$C in cores 1143 (red: Wang et al., 2004); 849 (blue: Mix et al., 1995); 846 (green: Shackleton et al., 1995). — the same δ$^{13}$C records filtered at 400-ky (bandpass = 2.5 Myr$^{-1}$) — eccentricity (grey, from Laskar et al., 2004) and filtered eccentricity (black).**



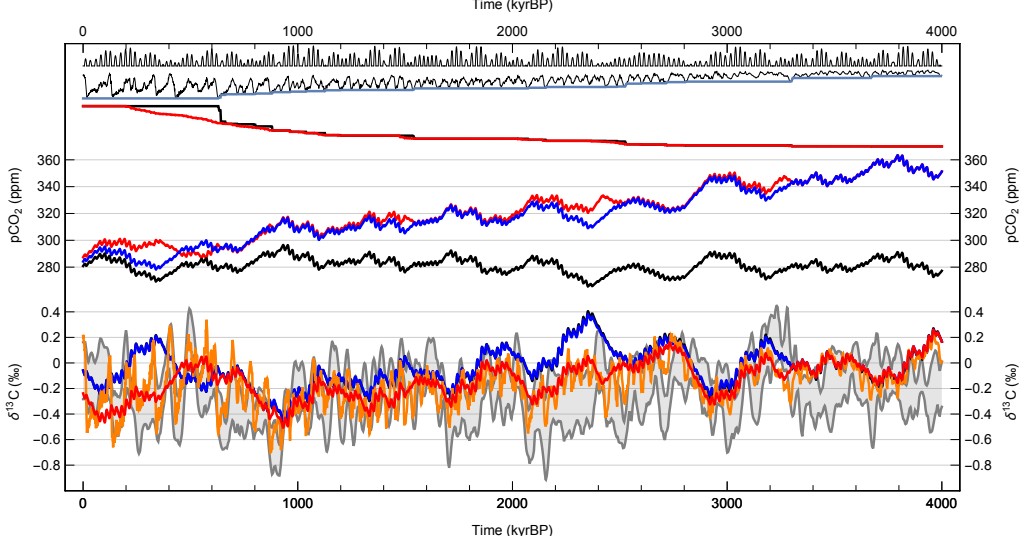

**Figure 2: From top to bottom: — Precessional forcing $F_0(t) = \text{Max}(0, -e\sin\omega)$ (black line) from Laskar et al. (2004). — Sea level curve LR04 (black line) from Lisiecki and Raymo (2005) used to compute the river incision $z_{MIN}$ defined as the previous sea level minima (blue line). — The geomorphological variable $s$ used from experiment $c$ (red lines) relaxed to its prescribed maximum value $s_{MAX} \sim z_{MIN}^3$ (black line). — Total carbon C rescaled as $pCO_2$ for experiments $a$ (black, precessional forcing only), $b$ (blue, idem, with a linear trend in carbon), and $c$ (red, using the geomorphological dynamics from equation (3)) — Carbon isotopic composition $\delta^{13}C$ for experiments $a$ (black), $b$ (blue), and $c$ (red). In orange, experiment $c$ with the addition of glacial-interglacial variability scaled on sea level. In grey, the min and max values of the $^{13}C$ records from Figure 1. In order to these results, we choose $t_C = 200$ kyr (Archer et al., 1997) or equivalently $V = 125$ TgC/yr. The trend (experiments $b$ and $c$) is set to $g = 1,2$ TgC/yr to induce a drift from about 350 ppm to about 280 ppm. The amplitude of the organic matter burial perturbation (experiments $a$, $b$ and $c$) is set to $a = 50$ TgC/yr. The filling rate of the sedimentary reservoir (experiment $c$) is set to $b = (160 \text{ kyr})^{-1}$. The model is integrated from an arbitrary initial condition at 5 MyrBP and the first 1 Myr is discarded.**



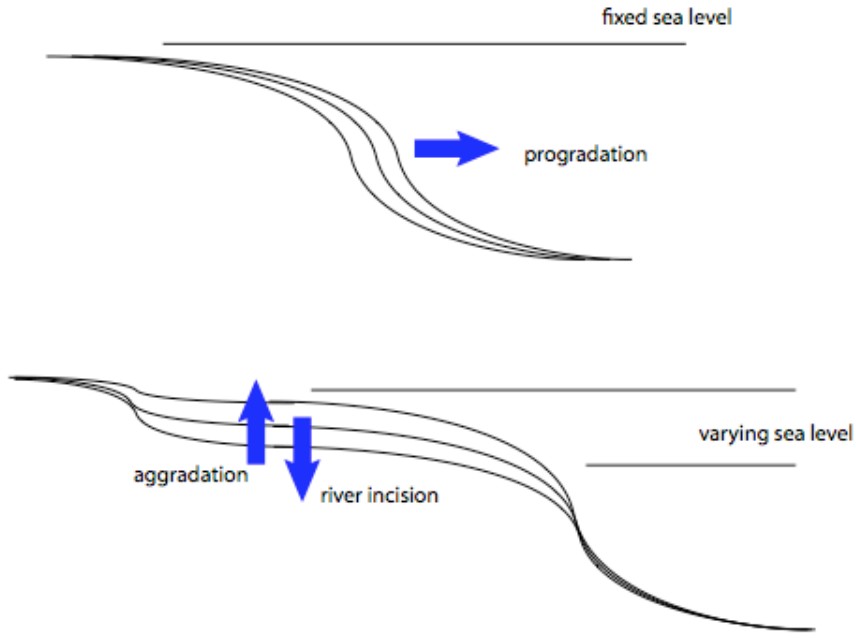

**Figure 3: Simple scheme of the two different geomorphological dynamics considered here : a/ With small sea level changes, we assume that the dominant sedimentary regime is progradation, with rather small organic carbon burial in coastal areas. The net effect of precessional forcing is (old) soil erosion, therefore a net transfer of carbon to the ocean-atmosphere ; b/ With large sea level changes during the late Quaternary, the dominant sedimentary regime can switch temporarily to aggradation just after major sea level drops and river incisions. During these transitory phases, the net effect of precessional forcing is reversed, with net organic carbon burial in river beds and river fans.**

