# Peer review of "The Plio-Pleistocene climatic evolution as a consequence of orbital forcing on the carbon cycle"

_Climate of the Past, 2017_

## Short Comment (SC1) · 8 Mar 2017

**Comment on *The Plio-Pleistocene climatic evolution as a consequence of orbital forcing on the carbon cycle* from D. Paillard, Clim. Past Discuss., doi:10.5194/cp-2017-3, 2017**

Peter Köhler

Alfred-Wegener-Institut Helmholtz-Zentrum für Polar-und Meeresforschung (AWI)

P.O. Box 12 01 61, 27515 Bremerhaven, Germany

email: Peter.Koehler@awi.de, Tel: +49 471 4831 1687

March 8, 2017

The paper of Paillard investigates the Plio-Pleistocene carbon cycle by setting up a conceptual model, consisting of differential equation for the carbon content of the atmosphere-ocean-biosphere $C$, the alkalinity of the ocean, $A$, and the stable carbon isotope values of $C$, $\delta^{13}C$. The analysis starts with the following equation for temporal changes in the carbon content of the system

$$\frac{\delta}{\delta t}C = V - B - D \tag{1}$$

with $V$ being the volcanic carbon input, $B$ the organic carbon burial, $D$ the oceanic carbonate deposition flux. Furthermore, the assumption that carbonate compensation will restore on multi-millennial years time scale the carbonate ion concentration is used. Here, the implicit used knowledge that alkalinity $A$ changes might be approximated after $\frac{\delta}{\delta t}A = W - 2D$ was used (but not mentioned explicitly) to final end with

$$\frac{\delta}{\delta t}C \;=\; 2(V - B) - W \tag{2}$$

$$\frac{\delta}{\delta t}(\delta^{13}C) \;=\; (V(-5 - \delta^{13}C) - B(-25 - \delta^{13}C)/C \tag{3}$$

with $W$ being the silicate weathering rate. According to the manuscript, the terms in brackets in Equation 3 are meant to be the following:

- $(-5 - \delta^{13}C)$: a volcanic source with constant isotopic signature of $-5‰$,

- $(-25 - \delta^{13}C)$: a constant $-25‰$ fractionation of organic matter with respect to the mean $\delta^{13}C$ of the considered system.

I find the conceptual idea how to understand the observed long-term changes in the carbon cycle very interesting. However, I have some fundamental comments to Equation 3 describing the evolution of the the carbon isotope of the system:

1. The term $(-5 - \delta^{13}C)$ in Equation (2) does not serve to describe the volcanic source with the constant isotopic signature of $-5\permil$ source as intended, but as some isotopic fractionation by $-5\permil$ with respect to the negative of the mean isotopic values $\delta^{13}C$ of the atmosphere-ocean-biosphere system.

2. The author decides to follow the initial Equation 1 when setting up the changes in the carbon isotopes. This approach is not wrong, but neglecting any impacts of the carbonate deposition flux $D$ on $\delta^{13}C$ might be too simple.

3. It is not clear to me, why in Equation 3 the isotopic signature of both fluxes $B$ and $V$ are described as a function of negative $\delta^{13}C$.

4. Changes in the isotopic value are always also depending on the content of the system. This implies that the differential equation has to be treated with care. In detail, one has to take into account, that when solving $\frac{\delta}{\delta t}(\delta^{13}C)$, one has to find a solution for $\frac{\delta}{\delta t}(C \cdot \delta^{13}C)$. Following the product rule, it follows:

$$\frac{\delta}{\delta t}(C \cdot \delta^{13}C) = C \cdot \frac{\delta}{\delta t}(\delta^{13}C) + \delta^{13}C \cdot \frac{\delta}{\delta t}(C) \tag{4}$$

Solving for $\frac{\delta}{\delta t}(\delta^{13}C)$ gives:

$$\frac{\delta}{\delta t}(\delta^{13}C) = \left(\frac{\delta}{\delta t}(C \cdot \delta^{13}C) - \delta^{13}C \cdot \frac{\delta}{\delta t}(C)\right) \cdot \frac{1}{C} \tag{5}$$

$$\frac{\delta}{\delta t}(\delta^{13}C) = \left(\text{RHS of DE} - \delta^{13}C \cdot \frac{\delta}{\delta t}(C)\right) \cdot \frac{1}{C} \tag{6}$$

The first term in equation (5), $\frac{\delta}{\delta t}(C \cdot \delta^{13}C)$ , is what is typically found on right-hand sides of differential equations (RHS of DE). The 2nd term in equations (5,6), $-\delta^{13}C \cdot \frac{\delta}{\delta t}(C)$, is probably small and might be negligible. However, its existence and any assumptions on neglecting it should in my view be mentioned for the sake of completeness.

5. When setting up differential equations for isotopes in the so-called $\delta$-notation one typically starts with the equation for the matter fluxes and multiplies each matter flux with the assumed isotopic signature of the flux, including any potential isotopic fractionation. With respect to the problem at hand (starting with equation 1) I end up with the following differential equation for $\delta^{13}C$:

$$\frac{\delta}{\delta t}(\delta^{13}C) = \left(V \cdot \delta^{13}C_V - B \cdot \delta^{13}C_B - D \cdot \delta^{13}C_D - \delta^{13}C \cdot \frac{\delta}{\delta t}(C)\right) \cdot \frac{1}{C} \tag{7}$$

According to the manuscript I get $\delta^{13}C_V = -5‰$ and $\delta^{13}C_B = \delta^{13}C - 25$. The argument that isotopic signature of the carbonate burial flux $D$ is identical to $\delta^{13}C$ of the system can be used to define $\delta^{13}C_D = \delta^{13}C$. Using the knowledge from the carbonate compensation that $D = W + B - V$ finally gives me a new equation for changes in the isotopic signature:

$$\frac{\delta}{\delta t}(\delta^{13}C) = \left( V \cdot (-5) - B \cdot (\delta^{13}C - 25) - (W + B - V) \cdot \delta^{13}C - \delta^{13}C \cdot \frac{\delta}{\delta t}(C) \right) \cdot \frac{1}{C} \quad (8)$$

I show in Figure 1 for the simplest scenario without long-term trend in $CO_2$ and the Amazon-like organic burial that both approaches (Paillard: Eq. 3; this comment: Eq. 8) lead to slightly different results, but they agree on large scale features. This comparison would therefore suggest, that the simplifications done in the setting up of the differential equation for the changes in the carbon isotopes by Paillard might be justified (even if I do not yet understand them in detail). However, I still believe that setting up the differential equation for the carbon isotope the way I describe above might be a way which is at least better to understand and easier to reproduce.

I like to finish with some more general comments:

1. Another simplification of the setup is the estimation of the change in atmospheric $CO_2$ mixing ratio by

$$CO_2 = 280 \cdot \left( \frac{C}{40,000 \text{ PgC}} \right)^2 \quad \text{(in ppm)}. \quad (9)$$

This equation was given without any further motivation. However, since all inputs of carbon to the system are given by volcanic $CO_2$ outgassing into the atmosphere, one might also evaluate the corresponding changes in atmospheric $CO_2$ concentration by the so-called airborne fraction, the fraction of injected carbon that stays in the atmosphere. With the given equation for $CO_2$ above (Eq. 9), this airborne fraction turns out to be around 3%. For example, a rise in C by 100 PgC, for example, would lead to a new $CO_2$ mixing ratio of 281.4 ppm. Following the well known relation of 1 ppm of $CO_2 = 2.12$ PgC, this rise in $CO_2$ by 1.4 ppm is similar to a rise in atmospheric carbon by nearly 3 PgC, thus 3% of the initial perturbation. The long-tail of the airborne fraction for potential future $CO_2$ emissions was recently investigated systematically with the GENIE Earth System Model. It was found (Equation S1 and Table S2 in Lord et al. (2016)) that the airborne fraction is around 5% and 1.6% on a timescale of $10^5$ and $10^6$ years, respectively. Thus, the so-far unmotivated assumption for $CO_2$ as given in Equation (7) above might be supported with such results but also illustrates, that variabilities faster than several $10^5$ years are not contained in this approximation of $CO_2$ given in Eq. 9.

2. Some of the assumptions are rather implicit and not supported with any further details or citations. The assumption that ocean alkalinity changes are approximated as changes in carbonate alkalinity by only considering variations in the carbonate ion concentration as $W - 2D$ was already mentioned above, and might find support in Zeebe and Wolf-Gladrow (2001). Furthermore, the assumption that the monsoon response to astronomical forcing as a simple function of the precessional forcing after $F_0 = \max(0, -e \cdot \sin(\omega))$ needs some backup from proxy reconstructions. References for the assumed isotopic signature of $-5\permil$ for volcanic outgassing $V$ and for the fractionation of $-25\permil$ in the organic burial flux $B$ would also be highly welcome.

3. The assumed long-term trend in weathering via the parameter $\gamma$ leads only to a decrease in carbon content and $CO_2$. Thus, to really mimic the multi-million decrease in $CO_2$ from 350 ppm 4 Myr ago to 280 ppm in the preindustrial time one needs also to increase the overall carbon content of the system at the beginning of the simulations. This is not mentioned. Otherwise $CO_2$ would start during scenarios which include this trend ($\gamma > 0$) at 280 ppm at the start of the simulations 4 Myr ago and decrease thereafter.

4. To reconstruct the carbon cycle in detail it would be helpful for the reader to be provided with the finally chosen parameter values.

**References**

Laskar, J., Robutel, P., Joutel, F., Gastineau, M., Correia, A. C. M., and Levrard, B.: A long term numerical solution for the insolation quantities of the Earth, Astronomy and Astrophysics, 428, 261–285, doi:10.1051/0004-6361:20041335, 2004.

Lord, N. S., Ridgwell, A., Thorne, M. C., and Lunt, D. J.: An impulse response function for the long tail of excess atmospheric CO2 in an Earth system model, Global Biogeochemical Cycles, 30, 2–17, doi:10.1002/2014GB005074, 2016.

Zeebe, R. E. and Wolf-Gladrow, D. A.: $CO_2$ in Seawater: Equilibrium, Kinetics, Isotopes, vol. 65 of *Elsevier Oceanography Book Series*, Elsevier Science Publishing, Amsterdam, The Netherlands, 2001.

[Figure]

Figure 1: Rebuilding the model of Paillard. A: Dimensionless orbital forcing function $F$ with an amplitude of 1 and a mean value of 0 based on Laskar et al. (2004). B: Change in atmospheric $CO_2$ concentration (following Eq. 9 of this comment) for the Amazon-like burial of organic carbon ($B = B_0 - aF(t)$, $a = 20$). No long-term trend in $CO_2$ is considered ($\gamma = 0$) and the carbon fluxes W, V, B are determined from the carbon turnover time of $\tau_C = 400$ kyr. C) Changes in $\delta^{13}C$ of the same Amazon-like burial scenario following either Paillard (Eq. 3) or this comment (Eq. 8).

---

## Referee Comment (RC1) · Anonymous Referee #1 · 14 Jun 2017

The contribution by D. Paillard explores the causes of the impact of orbital forcing on the long term evolution of the global carbon cycle. A simple model is built, accounting for the carbon, alkalinity, and carbon isotope mass balance. It is then forced by various mathematical functions (including periodic signals and long term trends). The author shows that the carbon cycle may have been controlled by the modulation of organic carbon burial in sediments, responding to orbital forcings.

This paper presents a clever method, and I think that such kind of conceptual study brings new valuable informations on how the carbon cycle operates in the latest Cenozoic.

I have only one point at this stage. It is related to equation 1a and 2b. One term is missing, but I'm not sure this will lead to major changes in the conclusions, invalidating

the study, or not. But this should be discussed.

Volcanism is not the only source of carbon at the million year timescale. The oxidation of petrogenic organic carbon exposed on the continental surfaces is releasing carbon to the ocean-atmosphere system. The rate is not well known, but it is probably of the same order of magnitude than the volcanic degassing (Blair et al., 2003, GCA). The C isotopic signature of this flux is quite different (around -25 permil) compared to volcanism. And its geological evolution depends on tectonic activity, physical erosion and continental runoff. The behavior of this source of carbon depends heavily on the geomorphic setting. Galy et al. (2007, Science) have shown in the specific case of the Himalaya that 50 to 70 % of the petrogenic carbon exposed in the Himalayas is being oxidized. In the case of the Amazon, it can be expected that most of this petrogenic organic carbon has been oxidized. Blair et al (2003, GCA) estimate a global flux of about 3 to 4 Tmol/yr of carbon released by this process. Note that this flux is included in all numerical model of the carbon cycle at the geological time scale (check Berner 2004 for instance).

This flux exerts a important control on the isotopic budget of carbon, owing to its negative signature. Incorporating it explicitly in the present model may change the results of the study. This additional source should be tested and its role discussed.

---

## Referee Comment (RC2) · Anonymous Referee #2 · 23 Jun 2017

This paper presents a nice conceptual model for interpreting changes in orbitally-paced variations in pCO2 and carbonate $\delta$13C through the past few million years. There are three components to the model. 1) A model of steady state carbon cycle fluxes and their isotopic composition. 2) A periodic term (linked to precession) modulating organic C burial. 3) A threshold term for the relationship between organic C burial and precession based on the global sea level curve.

For component (1), I see no error in the carbon cycle equations as written, but there are a few steps/assumptions that are not clearly articulated. Adding more details deriving each equation would make the paper easier to follow. In equation (1a) it is implicitly assumed that the weathering and volcanic fluxes can be lumped together (which is fine based on the assumption that both approximate the mantle isotopic value), though this

is not stated. (Otherwise the equation should be dC/dt = V + W - B – D). Next, I think it would be helpful to start with the full version of equation (2b):

d/dt($\delta$C*C) = V*$\delta$V - B*$\delta$B - D*$\delta$D

Then it would be more straightforward to see how the final version is obtained through the product rule and assumption that $\delta$c = $\delta$D as well as constant values of $\delta$V = -5‰ and $\delta$B = -25‰. This is particularly important because it is more typical to describe a constant fractionation of organic carbon with respect to $\delta$C, rather than a constant $\delta$B. On that note, adding an appropriate subscript to the $\delta$ notation (rather than writing as $\delta$13) would be helpful to differentiate between the $\delta$ values for each flux. Finally, there should be explanation of scaling between pCO2 and total C (namely, that the assumptions are being made that the ocean inventory of Ca2+ does not change and that the mass of carbon in the system is well-approximated by the ocean bicarbonate pool).

For component (2), it would be helpful to provide the chosen value for the scaling term a in equation (3) in the text and not just the caption to Fig. 2. Later in the paper, it is mentioned that a has to be of the same order as the equilibrium organic C burial flux, but the value in the caption is in fact double the equilibrium burial flux. There should also be a description of how this value was determined (presumably to get the right amplitude in the modeled $\delta$c)?

To me, component (3) is the most novel element of this conceptual model. This threshold term allows for a switch between two styles of periodic forcing of the organic carbon burial flux. In general, the periodic forcing reduces the value of B, except if the sedimentary reservoir is near to its maximum size, in which case periodic forcing switches to increasing the value of B. Again, the value of the scaling factor for the growth rate of the sedimentary reservoir, b, should be provided in the text, along with an explanation of how this value was determined. Next, what is the basis for setting the threshold condition at S < 0.85SMAX? The text notes that this threshold mechanism causes a

switch in organic carbon burial after significant sea level drops at 2.4-2.5 Myr and 0.35-0.65 Myr, but was the threshold set in order to provide this result? Also, in Figure 2, it is clear to see why the addition of this threshold term appreciably changes model behavior around 0.6 Myr, but not obviously earlier in the record. Maybe this is just hard to see because of the scale on the axes?

However, it does not seem that the conceptual model is particularly linked to the mechanism proposed (a shift between progradational to aggradational river systems). Paillard suggests in the introduction that "astronomical parameters are influencing climate through other mechanisms than the growth and decay of ice sheets," but it seems to me that what's been done is to link organic carbon burial to the growth and decay of ice sheets via the impact on sea level. This means the conceptual model is equally applicable to any process related to sea level that can drive a threshold response in organic carbon burial. This is not a flaw in the conceptual model, but parts of the text could be rewritten to emphasize that the geomorphological mechanism is only one possible physical interpretation of what the model actually describes.

Also, more discussion about the relationship between pCO2 and $\delta$13C cycles represented by this conceptual model would be welcome. Based on the introduction, I expected further explanation of phasing between simulated cycles and eccentricity. In particular, how well has the model accounted for a change in the nature of the 400 kyr $\delta$13C oscillation in the last million years? Also, why is the 100 kyr term added only to the modeled $\delta$13C and not pCO2? Perhaps add the eccentricity and filtered eccentricity to the same figure as the modeled curves.

Finally, in the results section of the text, comparison between blue and black curves in Figure 2 is cited as evidence for good agreement between model results and observations, but both these curves are model results.

---

## Author Comment (AC2) · 3 Jul 2017

First, I would like to thank the referee for his comments and encouragements. His main point concerns the role of petrogenic organic carbon, which represents a significant contributor to the long-term carbon cycle with also a significant role on the isotopic budget.

Indeed, my model considers only three sources and sinks of carbon : volcanic carbon (V) which is always a source ; carbonate precipitation (D) which always represents a net sink, though both dissolution and accumulation are considered through carbonate compensation; and finally organic carbon (B) which corresponds both to sinks (burial of recent organic matter), but also possibly to sources (oxidation of old organic car-

bon). Though this was not explicitly detailed in the manuscript, this last possibility (ie. a negative contribution to B, or "negative" burial) can be in part interpreted as a petrogenic organic carbon source. So implicitly, the model does already include petrogenic organic carbon. But, as explained by reviewer #1, this point needs to be discussed more precisely in a revised manuscript, since the negative contributions to B were only described as Âń net "old" soil erosion and remineralization Âż in the original submitted paper. Clearly, this was misleading.

As summarized by reviewer #2 (doi:10.5194/cp-2017-3-RC2) , the model is based first of all, on a rather standard steady-state equation for carbon. From the isotopic balance equation (2b), we deduce that the baseline (long-term) value B0 for all organic fluxes, including petrogenic ones, should be about 20% of the volcanic flux, that is B0 = V/5, in order to account for observed isotopic compositions. This baseline value B0 represents the sum of positive terms, mostly due to the burial of recent organic matter, but also negative ones that correspond to the oxidation of "old" soils and indeed "petrogenic" or "fossil" organic matter. As underlined by reviewer #1, the absolute magnitude of each term is currently not well constrained and positive and negative contributions to B0 are, individually, possibly comparable to V: indeed, if V is taken in the range of 40 to 175 TgC/yr
(Burton et al., 2013), the estimate for petrogenic organic carbon from Blair et al. (2003) [36 to 48 TgC/yr] corresponds to the lower range of V. This certainly needs to be explained in the revised manuscript.

Still the main point of the paper was not about the detailed steady state balance of the carbon system, but about its possible dynamics over the last 4 million years. For my model equations, only the net values of B0 (or B) are relevant. As explained by reviewer #1, the dynamics of petrogenic organic matter fluxes will depend on erosion and continental runoff. It will therefore contribute to the generic situation described in the manuscript, or "Amazon-like" situation, with enhanced organic carbon oxidation when precession maxima favours more precipitation and erosion. More precisely, when including precessional forcing through B = B0 - a F(t) with the numerical values B0 =

25 TgC/yr , a = 50 TgC/yr (see legend of Fig.2), then the "net burial" B does change sign through time, and becomes temporarily a carbon source when negative: it is then dominated by the oxidation of organic matter (soil, but also fossil or petrogenic. . .).

This will be discussed in more details in the revised manuscript.

---

## Author Comment (AC3) · 3 Jul 2017

Response to anonymous referee #2

First, I would like to thank the referee for his comments and support. He addresses several important technical points listed below and makes a more general remark, that my conceptual model is rather generic and could correspond to other geomorphological mechanisms than the one described in the manuscript. I believe that most of his comments can easily be addressed by a more explicit description of the model, its parameters, and its results, as explained below on a point-by-point basis (RC: the reviewer comment; AC: my response).

RC1 : For component (1), I see no error in the carbon cycle equations as written,

but there are a few steps/assumptions that are not clearly articulated. Adding more details deriving each equation would make the paper easier to follow. In equation (1a) it is implicitly assumed that the weathering and volcanic fluxes can be lumped together (which is fine based on the assumption that both approximate the mantle isotopic value), though this is not stated. (Otherwise the equation should be dC/dt = V + W - B – D).

AC1 : I somewhat disagree on this point. Silicate weathering W takes one CO2 molecule from the atmosphere (or more precisely H2CO3 from precipitation and runoff) and transforms it into HCO3- (through acido-basic reaction or proton exchange) in the river system and finally the ocean. When considering the oceanic carbon budget alone, W indeed adds one carbon in the ocean. But I am considering the "global" Earth surface budget (ocean + atmosphere) and therefore W has no net effect on C. Therefore W does not appear in equation (1a) for dC/dt. Its impact on the global carbon cycle arises only through the ocean alkalinity budget (dA/dt) and carbonate compensation, which leads to carbonate deposition D being directly linked to silicate weathering through D = W-V+B.

=> I will insist on the fact that C includes both the ocean and atmosphere, and better explain the underlying mechanisms.

RC2 : Next, I think it would be helpful to start with the full version of equation (2b): d/dt($\delta$C*C) = V*$\delta$V - B*$\delta$B - D*$\delta$D Then it would be more straightforward to see how the final version is obtained through the product rule and assumption that $\delta$c = $\delta$D as well as constant values of $\delta$V = -5‰ and $\delta$B = -25‰ This is particularly important because it is more typical to describe a constant fractionation of organic carbon with respect to $\delta$C, rather than a constant $\delta$B.

AC2 : This is indeed a good idea. This corresponds also to the remark from Peter Köhler (doi:10.5194/cp-2017-3-SC1) that the equations should be clarified, and the underlying assumptions should be more explicit.

[Figure]

=> I will add the derivation of equation (2b) and explicit choices for $\delta13V$, $\delta13B$, $\delta13D$.

RC3 : On that note, adding an appropriate subscript to the $\delta$ notation (rather than writing as $\delta13$) would be helpful to differentiate between the $\delta$ values for each flux.

AC3 : I will follow this suggestion and write the final equation (2b) as: $d\delta13C/dt = ( V(\delta13V-\delta13C) - B(\delta13B-\delta13C) )/C$

RC4 : Finally, there should be explanation of scaling between pCO2 and total C (namely, that the assumptions are being made that the ocean inventory of Ca2+ does not change and that the mass of carbon in the system is well-approximated by the ocean bicarbonate pool).

AC4 : This corresponds also to the remark from Peter Köhler. As explained in my response (doi:10.5194/cp-2017-3-AC1), this will be justified in the revised version.

RC5 : For component (2), it would be helpful to provide the chosen value for the scaling term a in equation (3) in the text and not just the caption to Fig. 2. Later in the paper, it is mentioned that a has to be of the same order as the equilibrium organic C burial flux, but the value in the caption is in fact double the equilibrium burial flux. There should also be a description of how this value was determined (presumably to get the right amplitude in the modeled $\delta$c)?

AC5 : I agree that a better discussion of parameter values could be included in the text, though these values are indeed determined empirically in order to get a qualitatively correct response. The amplitude a is indeed the double of the equilibrium flux B0 for the particular experiments shown on Fig.2. The comment in the text was slightly more generic (" the strength of the forcing a needs to be of the same order than the baseline value B0. This is a robust feature, which does not depend on model setting or parameters ").

=> I will add a short discussion on the choices made for a. I will rewrite the above sentence somewhat differently, as " when variations in B (or equivalently the parameter

a) are smaller than its baseline value B0, the model cannot reproduce the amplitude of $\delta$13C observed in marine benthic records ".

RC6 : To me, component (3) is the most novel element of this conceptual model. This thresh- old term allows for a switch between two styles of periodic forcing of the organic carbon burial flux. In general, the periodic forcing reduces the value of B, except if the sedimentary reservoir is near to its maximum size, in which case periodic forcing switches to increasing the value of B. . . Next, what is the basis for setting the threshold condition at S < 0.85 SMAX? The text notes that this threshold mechanism causes a switch in organic carbon burial after significant sea level drops at 2.4-2.5 Myr and 0.35-0.65 Myr, but was the threshold set in order to provide this result?

AC6 : I should certainly also be more explicit here. The "normal" (pre-Quaternary) sit- uation (progradation) is indeed when the periodic forcing reduces the value of B. Then the sedimentary reservoir S is typically at its maximum (we have S=Smax) as shown on Fig.2. But after every significant new sea-level drop (from the zmin "river incision" curve based on Lisiecki and Raymo, 2005), Smax = zmin3 increases significantly and the situation is switched to "aggradation". This first switch (switch ON) is not strongly dependent of the 0.85 threshold parameter, since a sea-level drop as small as about 5% will induce a sufficient increase in Smax (=15%) to trigger the change. But the switch back to normal (switch OFF) and therefore the duration of the "aggradation" phase, will depend more strongly on this threshold choice. In other words, concerning the two major "aggradation" phase discussed in the text (2.4-2.5 Myr and 0.35-0.65 Myr), their starts are directly linked to the significant sea level drops (at 2.5 Myr and 0.65 Myr): they are independent of the threshold value. But their duration is rather directly linked to this threshold value of 0.85 and also to the choice of parameter b. The choice of a different threshold value than 0.85 will consequently affect the amplitude of the differences between experiments (b) and (c), but not the timing of these differences.

=> This needs to be explained in the revised manuscript.

RC7 : Again, the value of the scaling factor for the growth rate of the sedimentary reservoir, b, should be provided in the text, along with an explanation of how this value was determined.

AC7 : I agree. And again, the value of b is a rather empirical choice. As explained above, its value will affect the duration of "aggradation" phases, and consequently the amplitude of the differences between experiments (b) and (c).

=> I will add a short discussion on the choices made for b.

RC8 : Also, in Figure 2, it is clear to see why the addition of this threshold term appreciably changes model behavior around 0.6 Myr, but not obviously earlier in the record. Maybe this is just hard to see because of the scale on the axes?

AC8 : There is indeed a significant change around 0.6 Myr that explains why the 400 kyr 13C cycles are disturbed at this time. There is also a significant change at about 2.4 MyrBP in the 13C results on Fig.2 (experiment (c): red curve) whereas the results without this mechanism (experiment (b): blue curve) the simulated 13C values are significantly out of the range of observed values. Interestingly, the switch model was designed to address the disturbed "400 kyr 13C cycles" of the last 1MyrBP. The better agreement with data at 2.4 MyrBP was not expected, and comes as a bonus.

=> I will clarify the role of the threshold mechanism when discussing results shown on Fig.2, and I will add a short comment on this last point in the conclusion.

RC9 : However, it does not seem that the conceptual model is particularly linked to the mech- anism proposed (a shift between progradational to aggradational river systems). Paillard suggests in the introduction that "astronomical parameters are influencing climate through other mechanisms than the growth and decay of ice sheets", but it seems to me that what's been done is to link organic carbon burial to the growth and decay of ice sheets via the impact on sea level. This means the conceptual model is equally applicable to any process related to sea level that can drive a threshold response in

organic carbon burial. This is not a flaw in the conceptual model, but parts of the text could be rewritten to emphasize that the geomorphological mechanism is only one possible physical interpretation of what the model actually describes.

AC9 : The first aim of this model is to link the observed 400-kyr 13C oscillations and the associated carbon cycle changes to the astronomical forcing, through the dynamics of organic matter burial. This is in general fully independent of sea level changes, except for the most recent Quaternary period. Since our knowledge of the carbon cycle is much more detailed over this recent period (pCO2 data, numerous 13C records, . . .), it is necessary to explain both the rather generic 400-kyr 13C oscillations observed during the Cenozoïc and beyond, but also why the Quaternary 13C oscillation look different and how this relates to observed pCO2 fluctuations. As explained in the introductory part of the paper, I am using a deductive line of thought. I certainly agree with the reviewer that the mechanism suggested here is probably not the only possible one. It is nevertheless (to my knowledge) the first one suggested so far that may explain both the recent past and the more remote one, in the same conceptual framework.

=> I will add a short comment on this last point in the conclusion, together with the following point (AC10 below).

RC10 : Also, more discussion about the relationship between pCO2 and $\delta$13C cycles rep- resented by this conceptual model would be welcome. Based on the introduction, I expected further explanation of phasing between simulated cycles and eccentricity. In particular, how well has the model accounted for a change in the nature of the 400 kyr $\delta$13C oscillation in the last million years?

AC10 : Indeed, it is probably important in the discussion to re-state the main objective of this model: reproducing not only the 400 kyr $\delta$13C oscillation seen during the pre-Quaternary, but also explaining why it is perturbed during the last million years, and to insist on the final $\delta$13C conclusion: assuming that this perturbation is caused by major sea level drops, as performed in this model, leads not only to a better agreement for

the $\delta$13C curves, but also explains several features of the CO2 changes.

=> This will be discussed in more details and more clearly re-stated in the conclusion.

RC11 : Also, why is the 100 kyr term added only to the modeled $\delta$13C and not pCO2?

AC11 : The 100-kyr term added to the 13C results (orange curve) is just an "ad-hoc" addition to improve the match with data, based on the (usually accepted) hypothesis that this 100-kyr oscillation in the 13C is liked to the varying size of the biosphere. There is no such data for the pCO2 over the last 4 million years, and there is no simple explanation for the observed pCO2 100-kyr cycles: adding this cycle a posteriori is therefore certainly not justified for pCO2. More importantly, the 100-kyr cycle is not the subject of this manuscript, so may be I should simply remove the orange curve to simplify the figure.

RC12 : Perhaps add the eccentricity and filtered eccentricity to the same figure as the modeled curves.

AC12 : Yes. This would indeed simplify the discussion of the results in terms of phasing, according to the above comments (RC10).

RC13 : Finally, in the results section of the text, comparison between blue and black curves in Figure 2 is cited as evidence for good agreement between model results and observations, but both these curves are model results.

AC13: This was a bad formulation in the original text. I meant that experiments a and b, with and without the long-term trend, (ie. the black and blue curves) were very similar in terms of 13C, and both were comparable to the data (the grey curves).

=> This sentence will be changed in the revised manuscript.
* * *

---

## Author Response (AR1)

**Response to anonymous referee #1**

First, I would like to thank the referee for his comments and encouragements. His main point concerns the role of petrogenic organic carbon, which represents a significant contributor to the long-term carbon cycle with also a significant role on the isotopic budget.

Indeed, my model considers only three sources and sinks of carbon : volcanic carbon (V) which is always a source ; carbonate precipitation (D) which always represents a net sink, though both dissolution and accumulation are considered through carbonate compensation; and finally organic carbon (B) which corresponds both to sinks (burial of recent organic matter), but also possibly to sources (oxidation of old organic carbon). Though this was not explicitly detailed in the manuscript, this last possibility (ie. a negative contribution to B, or "negative" burial) can be in part interpreted as a petrogenic organic carbon source. So implicitly, the model does already include petrogenic organic carbon. But, as explained by reviewer #1, this point needs to be discussed more precisely in a revised manuscript, since the negative contributions to B were only described as « net "old" soil erosion and remineralization » in the original submitted paper. Clearly, this was misleading.

As summarized by reviewer #2, the model is based first of all, on a rather standard steadystate equation for carbon. From the isotopic balance equation (2b), we deduce that the baseline (long-term) value  $B_0$  for all organic fluxes, including petrogenic ones, should be about 20% of the volcanic flux, that is  $B_0 = V/5$ , in order to account for observed isotopic compositions. This baseline value  $B_0$  represents the sum of positive terms, mostly due to the burial of recent organic matter, but also negative ones that correspond to the oxidation of "old" soils and indeed "petrogenic" or "fossil" organic matter. As underlined by reviewer #1, the absolute magnitude of each term is currently not well constrained and positive and negative contributions to  $B_0$  are, individually, possibly comparable to V: indeed, if V is taken in the range of 40 to 175 TgC/yr (Burton et al., 2013), the estimate for petrogenic organic carbon from Blair et al. (2003) [36 to 48 TgC/yr] corresponds to the lower range of V. This certainly needs to be explained in the revised manuscript.

=> I have now added the following paragraph in the revised manuscript:

(lines 107-113)

"It must be stressed that B stands for all organic carbon fluxes, whether they correspond to organic carbon burial (positive contributions to B) or to organic matter oxidation (negative contributions to B). If the long-term average equilibrium value  $B_0$  needs to be positive to account for the isotopic balance as shown above, this is not necessary always the case for the instantaneous values of B, as we will illustrate it below with the astronomical forcing. Indeed, B represents a sum of positive and negative terms whose individual absolute magnitudes are much larger than the long-term net value  $B_0$ . For instance, the oxidation of petrogenic organic carbon alone will contribute negatively to B, with a magnitude that may be as large as 40 TgC/yr (Blair et al., 2003)."

=> and also the mention of petrogenic organic carbon in the conclusion:

(lines 282-283)

"This model was built on the premises that changes in organic matter or petrogenic organic carbon fluxes are responsible for the 400-kyr oscillations observed in Cenozoïc 13C records"

Still the main point of the paper was not about the detailed steady state balance of the carbon system, but about its possible dynamics over the last 4 million years. For my model equations, only the net values of B0 (or B) are relevant. As explained by reviewer #1, the dynamics of petrogenic organic matter fluxes will depend on erosion and continental runoff. It will therefore contribute to the generic situation described in the manuscript, or "Amazon-like" situation, with enhanced organic carbon oxidation when precession maxima favours more precipitation and erosion. More precisely, when including precessional forcing through B = B0 - *a* F(t) with the numerical values B0 = 25 TgC/yr , *a* = 50 TgC/yr (see legend of Fig.2), then the "net burial" B does change sign through time, and becomes temporarily a carbon source when negative: it is then dominated by the oxidation of organic matter (soil, but also fossil or petrogenic...).

=> I also have added the following sentence:

(lines 146-148) "But this recent soil together with older soils and with petrogenic organic carbon (Galy et al., 2008) will be eroded and transported to the ocean through enhanced river discharges."

**Response to anonymous referee #2**

First, I would like to thank the referee for his comments and support. He addresses several important technical points listed below and makes a more general remark, that my conceptual model is rather generic and could correspond to other geomorphological mechanisms than the one described in the manuscript. I believe that most of his comments can easily be addressed by a more explicit description of the model, its parameters, and its results, as explained below on a point-by-point basis.

RC1: For component (1), I see no error in the carbon cycle equations as written, but there are a few steps/assumptions that are not clearly articulated. Adding more details deriving each equation would make the paper easier to follow. In equation (1a) it is implicitly assumed that the weathering and volcanic fluxes can be lumped together (which is fine based on the assumption that both approximate the mantle isotopic value), though this is not stated. (Otherwise the equation should be dC/dt = V + W - B - D).

AC1 : I somewhat disagree on this point. Silicate weathering W takes one  $CO_2$  molecule from the atmosphere (or more precisely  $H_2CO_3$  from precipitation and runoff) and transforms it into  $HCO_3^-$  (through acido-basic reaction or proton exchange) in the river system and finally the ocean. When considering the oceanic carbon budget alone, W indeed adds one carbon in the ocean. But I am considering the "global" Earth surface budget (ocean + atmosphere) and therefore W has no net effect on C. Therefore W does not appear in equation (1a) for dC/dt. Its impact on the global carbon cycle arises only through the ocean alkalinity budget (dA/dt) and carbonate compensation, which leads to carbonate deposition D being directly linked to silicate weathering through D = W-V+B.

=> I now insist on the fact that C includes both the ocean and atmosphere, and better explain the underlying mechanisms.

**(lines 91-95)**

Silicate weathering W takes one  $CO_2$  molecule from the atmosphere, or more precisely one  $H_2CO_3$  from precipitation and runoff, and transforms it into a  $HCO_3^-$  that finally reach the ocean. When considering the "global" Earth surface budget C which includes the ocean and atmosphere, W has therefore no direct effect on C and does not appear in equation (1a) for dC/dt, but only as a source of alkalinity in equation (1b).

*RC2* : Next, I think it would be helpful to start with the full version of equation (2b):  $d/dt(\delta C^*C) = V^*\delta V - B^*\delta B - D^*\delta D$

Then it would be more straightforward to see how the final version is obtained through the product rule and assumption that  $\delta c = \delta D$  as well as constant values of  $\delta V = -5\%$  and  $\delta B = -25\%$ . This is particularly important because it is more typical to describe a constant fractionation of organic carbon with respect to  $\delta C$ , rather than a constant  $\delta B$ .

AC2 : This is indeed a good idea. This corresponds also to the remark from Peter Köhler (doi:10.5194/cp-2017-3-SC1) that the equations should be clarified, and the underlying assumptions should be more explicit.

=> I now add the derivation of equation (2b) and explicit choices for  $\delta^{13}V$ ,  $\delta^{13}B$ ,  $\delta^{13}D$ .

(lines 114-125) The isotopic 13C budget can be written as:  $d/dt(C\,\delta^{13}C) = V\,\delta^{13}V - B\,\delta^{13}B - D\,\delta^{13}D$ where  $\delta^{13}C$  is the isotopic composition of ocean carbon,  $\delta^{13}V$  the isotopic composition of the volcanic carbon input,  $\delta^{13}B$  the isotopic composition of organic matter and  $\delta^{13}D$  the isotopic composition of marine carbonates. This can be re-written as:  $C \left( d\delta^{13}C/dt \right) + \left( dC/dt \right) \delta^{13}C = V \,\delta^{13}V - B \,\delta^{13}B - D \,\delta^{13}D$  $C (d\delta^{13}C/dt) = V \delta^{13}V - B \delta^{13}B - D \delta^{13}D - (V - B - D)\delta^{13}C$ or  $= V (\delta^{13}V - \delta^{13}C) - B (\delta^{13}B - \delta^{13}C) - D (\delta^{13}D - \delta^{13}C)$ If we neglect isotopic fractionation during carbonate precipitation (in other words,  $\delta^{13}D$ =  $\delta^{13}C$ ) and more generally during carbonate compensation, we finally obtain: (2b)  $d\delta^{13}C/dt = (V(\delta^{13}V \cdot \delta^{13}C) - B(\delta^{13}B \cdot \delta^{13}C))/C$ In the following we will assume a constant -5% volcanic source  $\delta^{13}V$ , as well as a constant -25% organic matter value  $\delta^{13}B$  (eq. Porcelli and Turekian, 2010).

*RC3*: On that note, adding an appropriate subscript to the  $\delta$  notation (rather than writing as  $\delta^{13}$ ) would be helpful to differentiate between the  $\delta$  values for each flux.

AC3 : I will follow this suggestion and write the final equation (2b) as:

 $d\delta^{13}C/dt = (V(\delta^{13}V - \delta^{13}C) - B(\delta^{13}B - \delta^{13}C))/C$

=> Equation (2b) is now written as: (line 123) (2b)  $d\delta^{13}C/dt = (V(\delta^{13}V-\delta^{13}C) - B(\delta^{13}B-\delta^{13}C))/C$

RC4: Finally, there should be explanation of scaling between pCO2 and total C (namely, that the assumptions are being made that the ocean inventory of Ca2+ does not change and that the mass of carbon in the system is well-approximated by the ocean bicarbonate pool).

AC4 : This corresponds also to the remark from Peter Köhler. As explained in my response (doi:10.5194/cp-2017-3-AC1), this will be justified in the revised version.

=> The scaling of pCO2 is now explained in the text as followed:

(lines 126-130)

Indeed, if we assume, to first order, that C may represent the carbon content of a wellmixed ocean, then from chemical equilibrium  $pCO_2$  should be proportional to  $[HCO_3^-]^2/$  $[CO_3^{2-}]$ . After carbonate compensation (ie. assuming that  $[CO_3^{2-}]$  remains constant) and considering that C is dominated by bicarbonates  $[HCO_3^-]$  under standard pH conditions, we end up with the approximate scaling that  $pCO_2$  varies roughly as  $C^2$

(lines 94-96)

if we assume that the oceanic calcium concentration does not change significantly over the last few millions of years, carbonate compensation will restore the oceanic carbonate content. RC5 : For component (2), it would be helpful to provide the chosen value for the scaling term a in equation (3) in the text and not just the caption to Fig. 2. Later in the paper, it is mentioned that a has to be of the same order as the equilibrium organic C burial flux, but the value in the caption is in fact double the equilibrium burial flux. There should also be a description of how this value was determined (presumably to get the right amplitude in the modeled  $\delta c$ )?

AC5 : I agree that a better discussion of parameter values could be included in the text, though these values are indeed determined empirically in order to get a qualitatively correct response. The amplitude *a* is indeed the double of the equilibrium flux  $B_0$  for the particular experiments shown on Fig.2. The comment in the text was slightly more generic (*« the strength of the forcing a needs to be of the same order than the baseline value B*0. *This is a robust feature, which does not depend on model setting or parameters »*).

I will add a short discussion on the choices made for *a*. I will rewrite the above sentence somewhat differently, as *«when variations in B (or equivalently the parameter a) are smaller than its baseline value B0, the model cannot reproduce the amplitude of \delta^{13}C observed in marine benthic records ».*

 $\Rightarrow$  I have now added a short discussion on the choices made for *a*, and modified the ambiguous sentence.

(lines 161-162) The value of the parameter a is chosen in order to obtain approximately the correct amplitude for these simulated 400-kyr oscillations (a = 50 TgC/yr).

(lines 217-218) When variations in *B*, as determined by parameter *a*, are smaller than the baseline value  $B_0$ , the model cannot reproduce the oceanic amplitude of  $\delta^{13}C$  observed in marine benthic records

RC6 : To me, component (3) is the most novel element of this conceptual model. This threshold term allows for a switch between two styles of periodic forcing of the organic carbon burial flux. In general, the periodic forcing reduces the value of B, except if the sedimentary reservoir is near to its maximum size, in which case periodic forcing switches to increasing the value of B...Next, what is the basis for setting the threshold condition at S < 0.85 SMAX? The text notes that this threshold mechanism causes a switch in organic carbon burial after significant sea level drops at 2.4-2.5 Myr and 0.35-0.65 Myr, but was the threshold set in order to provide this result?

AC6 : I should certainly also be more explicit here. The "normal" (pre-Quaternary) situation ("progradation") is indeed when the periodic forcing reduces the value of B. Then the sedimentary reservoir S is typically at its maximum (we have S=Smax) as shown on Fig.2. But after every significant new sea-level drop (from the zmin "river incision" curve based on Lisiecki and Raymo, 2005), Smax =  $zmin^3$  increases significantly and the situation is switched to "aggradation". This first switch ("switch ON") is not strongly dependent of the 0.85 threshold parameter, since a sea-level drop as small as about 5% will induce a sufficient increase in Smax (=15%) to trigger the change. But the switch back to

normal ("switch OFF") and therefore the duration of the "aggradation" phase, will depend more strongly on this threshold choice. In other words, concerning the two major "aggradation" phase discussed in the text (2.4-2.5 Myr and 0.35-0.65 Myr), their starts are directly linked to the significant sea level drops (at 2.5 Myr and 0.65 Myr): they are independent of the threshold value. But their duration is rather directly linked to this threshold value of 0.85 and also to the choice of parameter b. The choice of a different threshold value than 0.85 will consequently affect the amplitude of the differences between experiments (b) and (c), but not the timing of these differences.

=> This is now explained in the revised manuscript.

(lines 193-196)

After a short transient period, this reservoir remains therefore equal to this maximum value  $S_{MAX}$  in the absence of major sea level drops, as during the pre-Quaternary period. In contrast, for each significant sea level drop,  $S_{MAX}$  increases abruptly and we start a new transient phase whose duration is linked to parameter b.

(lines 206-207)

The start of these transient periods is directly linked to sea level drops, according to the LR04 forcing, while the duration of these transients is linked both to the 0.85  $S_{MAX}$  threshold and the b parameter.

*RC7* : Again, the value of the scaling factor for the growth rate of the sedimentary reservoir, *b*, should be provided in the text, along with an explanation of how this value was determined.

AC7 : I agree. And again, the value of b is a rather empirical choice. As explained above, its value will affect the duration of "aggradation" phases, and consequently the amplitude of the differences between experiments (b) and (c).

 $\Rightarrow$  I now present the rational behind choices made for *b*.

(lines 207-208)

... whose values are chosen to qualitatively better match the  $\delta^{13}C$  data. For the results show on Fig.2,  $b = (160 \text{ kyr})^{-1}$ .

RC8 : Also, in Figure 2, it is clear to see why the addition of this threshold term appreciably changes model behavior around 0.6 Myr, but not obviously earlier in the record. Maybe this is just hard to see because of the scale on the axes?

AC8 : There is indeed a significant change around 0.6 Myr that explains why the "400 kyr 13C cycles" are disturbed at this time. There is also a significant change at about 2.4 MyrBP in the 13C results on Fig.2 (experiment (c): red curve) whereas the results without this mechanism (experiment (b): blue curve) the simulated 13C values are significantly out of the range of observed values. Interestingly, the "switch" model was designed to address the disturbed "400 kyr 13C cycles" of the last 1 MyrBP. The better agreement with data at 2.4 MyrBP was not expected, and comes as a bonus.

=> I now clarify the role of the threshold mechanism when discussing results shown on Fig.2, and I have added a short comment on this last point in the discussion.

*(lines 210-211) "...as illustrated by the difference between the blue and red curves on Fig.2".*

(lines 250-251)

This mechanism also allows for simulated marine  $\delta^{13}C$  in better agreement with data at about 2.4 MyrBP.

=> but this comment also probably arises because, if the mode-switch is rather clear on Fig.2 when looking at the 13C results, it is less so on the geomorphological variables s and Smax. I have therefore added an orange shading on top of these curves, corresponding to the transient aggradation regimes (S < 0.85 SMAX).

(added shading in Fig. 2)

RC9 : However, it does not seem that the conceptual model is particularly linked to the mechanism proposed (a shift between progradational to aggradational river systems). Paillard suggests in the introduction that "astronomical parameters are influencing climate through other mechanisms than the growth and decay of ice sheets," but it seems to me that what's been done is to link organic carbon burial to the growth and decay of ice sheets via the impact on sea level. This means the conceptual model is equally applicable to any process related to sea level that can drive a threshold response in organic carbon burial. This is not a flaw in the conceptual model, but parts of the text could be rewritten to emphasize that the geomorphological mechanism is only one possible physical interpretation of what the model actually describes.

AC9 : The first aim of this model is to link the observed 400-kyr 13C oscillations and the associated carbon cycle changes to the astronomical forcing, through the dynamics of organic matter burial. This is in general fully independent of sea level changes, except for the most recent Quaternary period. Since our knowledge of the carbon cycle is much more detailed over this recent period (pCO2 data, numerous 13C records, ...), it is necessary to explain both the rather generic 400-kyr 13C oscillations observed during the Cenozoïc and beyond, but also why the Quaternary 13C oscillation look different and how this relates to observed pCO2 fluctuations. As explained in the introductory part of the paper, I am using a deductive line of thought. I certainly agree with the reviewer that the mechanism suggested here is probably not the only possible one. It is nevertheless (to my knowledge) the first one suggested so far that may explain both the recent past and the more remote one, in the same conceptual framework.

=> I have added the following paragraph in the conclusion

**(lines 282-288)**

"This model was built on the premises that changes in organic matter or petrogenic organic carbon fluxes are responsible for the 400-kyr oscillations observed in Cenozoïc 13C records, and that the large sea-level variations occurring during the Quaternary are strongly affecting this process. Continental margins and sedimentary fans are a very likely key component, as illustrated by our simple conceptual model. But obviously, many complex processes are involved in the interactions between organic matter burial or oxidation, monsoons and sea-level changes. The geomorphological mechanism described here is one possibility which allows, for the first time, to account both for the persistent 400-kyr oscillation observed in 13C records during the Cenozoïc, but also for its change during the last million years"

RC10 : Also, more discussion about the relationship between pCO2 and  $\delta 13C$  cycles represented by this conceptual model would be welcome. Based on the introduction, I expected further explanation of phasing between simulated cycles and eccentricity. In particular, how well has the model accounted for a change in the nature of the 400 kyr  $\delta^{13}C$  oscillation in the last million years?

AC10 : Indeed, it is probably important in the discussion to re-state the main objective of this model: reproducing not only the 400 kyr  $\delta^{13}$ C oscillation seen during the pre-Quaternary, but also explaining why it is perturbed during the last million years, and to insist on the final  $\delta^{13}$ C conclusion: assuming that this perturbation is caused by major sea level drops, as performed in this model, leads not only to a better agreement for the  $\delta^{13}$ C curves, but also explains several features of the CO2 changes.

=> This is now discussed in more details in the discussion,

**(lines 240-251)**

"According to this mechanism, in the ordinary sedimentary regime (progradation), we obtain changes in the carbon cycle with  $pCO_2$  maxima and  $\delta^{13}C$  minima associated directly to eccentricity maxima. This is indeed consistent with long Cenozoïc records (eg. Pälike et al, 2006).

When we allow for changes in the sedimentary regime triggered by sea level changes, the model can also reproduce more peculiar features. Indeed, up to now it has been difficult to explain the last two long-term cycles observed in the marine  $\delta^{13}C$ , each being approximately 500 kyr-long, with a maximum now ( $\delta^{13}$ Cmax-I), a well-marked maximum at about 500 kyr BP ( $\delta^{13}$ Cmax-II) and a previous one around 1000 or 1100 kyr BP ( $\delta^{13}$ Cmax-II). In the model described here, these two long oscillations are generated from the eccentricity forcing, but with an abrupt switch to aggradation mode at about 620 kyrBP caused by the sea level drop at MIS 16. This switch reverses the phase of the 400-kyr carbon oscillation during a few hundred thousands of years. Interestingly, this also induces a slight minimum in the carbon (or pCO2) results, consistent with the observed low pCO2 values observed in the Antarctic ice core around 600-700 kyrBP. This mechanism also allows for simulated marine  $\delta^{13}C$  in better agreement with data at about 2.4 MyrBP. »

**RC11* : Also, why is the 100 kyr term added only to the modeled $\delta 13C$ and not pCO2?**

AC11 : The 100-kyr term added to the 13C results (orange curve) is just an "ad-hoc" addition to improve the match with data, based on the (usually accepted) hypothesis that this 100-kyr oscillation in the 13C is linked to the varying size of the biosphere. There is no such data for the pCO2 over the last 4 million years, and there is no simple explanation for the observed pCO2 100-kyr cycles: adding this cycle *a posteriori* is therefore certainly not justified for pCO2. More importantly, the 100-kyr cycle is not the subject of this manuscript, so may be I should simply remove the orange curve to simplify the figure.

=> I have removed the 100-kyr "ad-hoc" addition on the 13C results (orange curve) and the corresponding paragraph in the main text.

(removed lines 215...)

RC12: Perhaps add the eccentricity and filtered eccentricity to the same figure as the modeled curves.

AC12 : Yes. This would indeed simplify the discussion of the results in terms of phasing, according to the above comments (RC10).

=> I have added the filtered eccentricity at the bottom of Fig.2, as well as the filtered  ${}^{13}C$  data and model results. This new figure setting corresponds also better to the data shown on Fig.1, in order to simplify the understanding of the figure.

(added curves in Fig. 2)

*RC13* : Finally, in the results section of the text, comparison between blue and black curves in Figure 2 is cited as evidence for good agreement between model results and observations, but both these curves are model results.

AC13: This was a bad formulation in the original text. I meant that experiments a and b, with and without the long-term trend, (ie. the black and blue curves) were very similar in terms of 13C, and both were comparable to the data (the grey curves).

=> This is now changed in the text as:

(lines 163-165)

More specifically, the  $\delta^{13}C$  black and blue simulated curves are superimposed and almost undistinguishable, since the linear trend added to the carbon cycle has almost no impact on the  $\delta^{13}C$ . They are both most of the time within the range of observed values (gray curves).

**Response to Peter Köhler**

First, I would like to thank Peter Köhler for providing these thoughtful comments and for rebuilding and reproducing my model results. I acknowledge that the model description and the presentation of some equations or parameters was sometimes not explicit enough in the submitted manuscript. I therefore want to clarify some points below.

**1. The 13C equation.**

The main point raised by Peter Köhler concerns the  $^{13}$ C equation. It turns out that we are both using (almost) the same equation, but just written differently. The only true difference stands in the organic matter fractionation : while Peter Köhler uses a constant fractionation with respect to the environment, I implicitly considered an organic matter sink with a constant isotopic signature of -25‰.

More precisely, equation (8) from Peter Köhler reads :

$$\frac{d}{dt}(\delta^{13}C) = \frac{1}{C} \left[ V(-5) - B(\delta^{13}C - 25) - (W + B - V)\delta^{13}C - \delta^{13}C \cdot \frac{dC}{dt} \right]$$

When substituting the last term using the equation for dC/dt :

$$\frac{d}{dt}C = V - B - D$$

we get :

$$\frac{d}{dt}(\delta^{13}C) = \frac{1}{C}[V(-5) - B(\delta^{13}C - 25) - (W + B - V)\delta^{13}C - \delta^{13}C.(V - B - D)]$$

And after simplification, and using D = W+B-V, this leads to :

$$\frac{d}{dt}(\delta^{13}C) = \frac{1}{C}[V(-5 - \delta^{13}C) - B(-25)]$$

As mentioned above, this is very similar to my equation (2b) :

$$\frac{d}{dt}(\delta^{13}C) = \frac{1}{C}[V(-5 - \delta^{13}C) - B(-25 - \delta^{13}C)]$$

the only difference being that, implicitly, I used a constant organic matter sink of -25‰. Since the 13C of carbonates remains close to 0‰, these different choices lead to a very small difference in the numerical experiments, as demonstrated by Peter Köhler. In any case, this point should be clarified in a revised manuscript.

=> Following also the advice of Rev.#2 (RC2), I now add the explicit derivation of equation (2b) and explicit choices for  $\delta^{13}V$ ,  $\delta^{13}B$ ,  $\delta^{13}D$ .

(lines 114-125) The isotopic 13C budget can be written as:  $d/dt(C \ \delta^{13}C) = V \ \delta^{13}V - B \ \delta^{13}B - D \ \delta^{13}D$ where  $\delta^{13}C$  is the isotopic composition of ocean carbon,  $\delta^{13}V$  the isotopic composition of the volcanic carbon input,  $\delta^{13}B$  the isotopic composition of organic matter and  $\delta^{13}D$  the isotopic composition of marine carbonates. This can be re-written as:  $C (d\delta^{13}C/dt) + (dC/dt) \ \delta^{13}C = V \ \delta^{13}V - B \ \delta^{13}B - D \ \delta^{13}D$  or  $C(d\delta^{13}C/dt) = V \delta^{13}V - B \delta^{13}B - D \delta^{13}D - (V - B - D)\delta^{13}C$  $= V(\delta^{13}V - \delta^{13}C) - B(\delta^{13}B - \delta^{13}C) - D(\delta^{13}D - \delta^{13}C)$ If we neglect isotopic fractionation during carbonate precipitation (in other words,  $\delta^{13}D$  $= \delta^{13}C$ ) and more generally during carbonate compensation, we finally obtain: (2b)  $d\delta^{13}C/dt = (V(\delta^{13}V - \delta^{13}C) - B(\delta^{13}B - \delta^{13}C))/C$ In the following we will assume a constant -5‰ volcanic source  $\delta^{13}V$ , as well as a constant -25‰ organic matter value  $\delta^{13}B$  (eg. Porcelli and Turekian, 2010).

2. The pCO2 scaling equation.

In the manuscript, C represent the total carbon content at the « Earth surface », which means mostly the ocean reservoir, plus a minor contribution from the biosphere and atmosphere. I used a simple scaling to translate these changes in carbon content C (expressed in GtC) in terms of atmospheric  $pCO_2$  (in ppm) :

$$pCO_2 = 280 \left(\frac{C}{40,000}\right)^2$$

As explained by Peter Köhler, this might be supported by model experiments for long time scales, but this lacks some justification in the manuscript. Such a scaling can be obtained when considering that C represent the carbon content of a well-mixed ocean. Then, from chemical equilibrium, we obtain :

$$pCO_2 = k \frac{[HCO_3^-]^2}{[CO_3^{2-}]}$$

where the constant *k* includes the solubility of CO2, and the first and second dissociation constants of carbonate and bicarbonate ions. When considering only the long time scale response, we can assume that carbonate compensation will restore  $[CO_3^{2-}]$  to a constant initial value. Furthermore, under standard oceanic pH conditions, bicarbonate ions  $[HCO_3^{-}]$  represent about 90% of the total carbon content C. If we assume, to first order, that  $C \approx [HCO_3^{-}]$ , then the above equation means that pCO2 should, on long time scales, increase approximately as the square of C. Though this is certainly a rough approximation, it is sufficient to provide a reasonable magnitude of the implied pCO2 changes associated with this simple model.

Again, this point should be clarified in a revised manuscript

=> Following also the advice of Rev.#2 (RC4), the scaling of pCO2 is now explained in the text as followed:

(lines 126-130) Indeed, if we assume, to first order, that C may represent the carbon content of a wellmixed ocean, then from chemical equilibrium  $pCO_2$  should be proportional to  $[HCO_3^-]^2/$  $[CO_3^{2-}]$ . After carbonate compensation (ie. assuming that  $[CO_3^{2-}]$  remains constant) and considering that C is dominated by bicarbonates  $[HCO_3^-]$  under standard pH conditions, we end up with the approximate scaling that  $pCO_2$  varies roughly as  $C^2$

(lines 94-96)

if we assume that the oceanic calcium concentration does not change significantly over the last few millions of years, carbonate compensation will restore the oceanic carbonate content. 3. I indeed also used the rather implicit assumptions that ocean alkalinity is approximated by carbonate alkalinity, therefore equation (1b) in the manuscript. This could be discussed a bit more in the manuscript, though it is quite a classical approximation.

=> addition (line 90) ...assuming that alkalinity is dominated by carbonate alkalinity.

Concerning the choice of precessional forcing  $F_0(t) = \max(0, -e \sin\omega)$ , I am not sure that any proxy comparison would either backup or dismiss such a choice. Furthermore, there is little hope to find any proxy for global organic carbon preservation, since individual proxies of preservation are often very dependent of the local or regional context. The choice of this forcing is simply based on two premises: 1 – monsoon are primarily driven by precession, something demonstrated by paleoclimatic data and simulated by climate model. 2 – the conceptual model needs a rectifying mechanism to reproduce the envelope of precession, something consistent with the averaged values of river sedimentary carbon discharges being largely dominated by the largest or extreme events. The expression above is the simplest possible choice along these lines.

The isotopic signatures used for volcanic outgassing (-5‰) and for buried organic matter (-25‰) are rather standard values used in geochemical textbooks and treatises. For instance:

Porcelli, D. and Turekian, K.K., The History of Planetary Degassing as Recorded by Noble Gases, §6.6.1 in Readings from the Treatise on Geochemistry, edited by Holland, H.D. and Turekian, K.K., (2010).

These numbers are somewhat conventional with actual measurements varying from about -1‰ to -8‰ for volcanoes or mid-ocean ridges outgassing, depending on location. Similarly, -25‰ is a conventional value for organic matter  $\delta^{13}$ C used for instance as a normalization for reporting 14C activities, while actual values vary from roughly -10‰ to -30‰ depending on organic materials.

=> reference added

(lines 124-125) In the following we will assume a constant -5% volcanic source  $\delta^{13}$ V, as well as a constant -25% organic matter value  $\delta^{13}$ B (eg. Porcelli and Turekian, 2010).

4. There is no need to specify the carbon content of the model, since it is explicitly computed by the equations. As mentioned in Figure caption 2, the model is integrated from an arbitrary condition (that is carbon content, and isotopic value) at 5 MyrBP and the first 1 Myr is discarded, since it correspond to the transient part of the simulation.

=> This is now also explicitly mentioned in the text.

(lines 133-134) The model is integrated from an arbitrary initial condition at 5 MyrBP and the first 1 Myr is discarded. 5. I believe all parameter values are given in Figure caption 2, but there has been an unfortunate typesetting change from greek to latin alphabet. This also needs to be corrected in a revised manuscript.

=> typesetting has been checked => parameter values are now given also in the main text (cf. Rev.#2, RC5 & RC7). (line 161) (a = 50 TgC/yr). (line 207) b = (160 kyr)-1. (line 132) γ set to 1,2 TgC/yr

[revised manuscript text omitted]

Didier Paillard 8/8/y 11:59 Commentaire [1]: Peter Köhler's comment #3

runoff, and transforms it into a HCO3- that finally reach the ocean. When considering the "global" Earth surface budget C which includes the ocean and atmosphere, W has therefore no direct effect on C and does not appear in equation (1a) for dC/dt, but only as a source of alkalinity in equation (1b). On time scales larger than several millennia, if we assume that the

95 oceanic calcium concentration does not change significantly over the last few millions of years, carbonate compensation will restore the oceanic carbonate content. Therefore, to first order, we can write :

$$d[CO_3^{2-}]/dt = d(A-C)/dt = 0 = W - D - V + B.$$

Solving for D, this leads to the long-term evolution equations for carbon:

 $(2a) \qquad \qquad dC/dt = 2(V-B) - W$

For simplicity, we will assume that the main stabilizer of the carbon system is the silicate weathering, with a fixed relaxation time  $\tau_C$ , ie. W = C/ $\tau_C$ . Solving the present-day equilibrium with  $\delta^{13}_{Eq} = 0\%$  as a typical value for carbonates, we easily deduce typical equilibrium values for the fluxes : B0 = V/5 ; CEq = (8/5)  $\tau_C V = 40,000$  PgC. If we assume a relaxation time  $\tau_C$  of 200 kyr (Archer et al., 1997), we obtain V = (5/8) CEq/ $\tau_C = 125$  TgC/yr and B0 = 25 TgC/yr. For a larger value  $\tau_C = 400$ kyr (Archer et al. 2005), we would get V = 62 TgC/yr. There is no consensus on the actual total carbon emissions from volcanism (including all aerial and submarine sources), but these values for V (or  $\tau_C$ ) span more or less the range of current

estimates from about 40 to 175 TgC/yr (Burton et al, 2013). It must be stressed that B stands for all organic carbon fluxes, whether they correspond to organic carbon burial (positive contributions to B) or to organic matter oxidation (negative contributions to B). If the long-term average equilibrium value B0 needs to be positive to account for the isotopic balance as shown above, this is not necessary always the case for the

110 instantaneous values of B, as we will illustrate it below with the astronomical forcing. Indeed, B represents a sum of positive and negative terms whose individual absolute magnitudes are much larger than the long-term net value B0. For instance, the oxidation of petrogenic organic carbon alone will contribute negatively to B, with a magnitude that may be as large as 40 TgC/yr (Blair et al., 2003).

The isotopic 13C budget can be written as:

120

115  $d/dt(C \ \delta^{13}C) = V \ \delta^{13}V - B \ \delta^{13}B - D \ \delta^{13}D$

where  $\delta^{13}C$  is the isotopic composition of ocean carbon,  $\delta^{13}V$  the isotopic composition of the volcanic carbon input,  $\delta^{13}B$  the isotopic composition of organic matter and  $\delta^{13}D$  the isotopic composition of marine carbonates. This can be re-written as:  $C (d\delta^{13}C/dt) + (dC/dt) \delta^{13}C = V \delta^{13}V - B \delta^{13}B - D \delta^{13}D$

or
$$C (d\delta^{13}C/dt) = V \delta^{13}V - B \delta^{13}B - D \delta^{13}D - (V - B - D)\delta^{13}C$$

= V (
$$\delta^{13}$$
V- $\delta^{13}$ C) - B ( $\delta^{13}$ B- $\delta^{13}$ C) - D ( $\delta^{13}$ D -  $\delta^{13}$ C)

If we neglect isotopic fractionation during carbonate precipitation (in other words,  $\delta^{13}D = \delta^{13}C$ ) and more generally during carbonate compensation, we finally obtain:

4

(2b)  $d\delta^{13}C/dt = (V(\delta^{13}V - \delta^{13}C) - B(\delta^{13}B - \delta^{13}C))/C$

Didier Paillard 8/8/y 11:42 Commentaire [2]: Rev.#2, RC1

Didier Paillard 8/8/y 13:26 Commentaire [3]: Rev.#2, RC4

Commentaire [4]: Rev.#1

Didier Paillard 8/8/y 11:54 Commentaire [5]: Rev.#2, RC3 In the following we will assume a constant -5% volcanic source  $\delta^{13}V$ , as well as a constant -25‰ organic matter value  $\delta^{13}B$

125 (eg. Porcelli and Turekian, 2010).

In order to translate the total carbon content C into an equivalent pCO2 level, we will use a simple scaling. Indeed, if we assume, to first order, that C may represent the carbon content of a well-mixed ocean, then from chemical equilibrium pCO2 should be proportional to  $[HCO_3^-]^2/[CO_3^{2-}]$ . After carbonate compensation (i.e. assuming that  $[CO_3^{2-}]$  remains constant) and considering that C is dominated by bicarbonates  $[HCO_3^-]$  under standard pH conditions, we end up with the approximate

- 130 scaling that pCO2 varies roughly as C2, or pCO2 = 280 (C/40,000)2 (in ppm). To reproduce a multi-million year trend, we need to add one explicitly in the weathering relaxation:  $W = C/\tau_C = (\Delta C + C_{Eq} - \gamma t)/\tau_C$ , with the coefficient  $\gamma$  set to 1,2 TgC/yr to obtain the desired pCO2 levels at the start of the simulation, ie. about 350 ppm at 4 MyrBP, according to current estimates (Bartoli et al., 2011; Seki et al., 2010). The model is integrated from an arbitrary initial condition at 5 MyrBP and the first 1 Myr is discarded.
- 135 In the following, we are describing how carbon burial B should vary with monsoons, and what consequences these variations have on the total carbon content C as well as on carbonate isotopes  $\delta^{13}$ C. In order to represent the monsoon's response to astronomical forcing, we introduce a simple truncation of the precessional forcing:

 $F_0(t) = \max(0, -e\sin\omega)$

where e is the eccentricity and  $\omega$  the climatic precession.

140 Indeed, soil erosion or sediment transport are dominated by intense events, not by the average climate. Such a non-linear response can be mimicked in a simple way by the above expression that accounts only for positive monsoonal forcing, not for negative one. Consequently, the model will be influenced by the amplitude modulation of the precessional forcing, ie. the eccentricity. To avoid useless parameters, we furthermore introduce the normalization:

 $F = F_0 / Max(F_0) - \langle F_0 / Max(F_0) \rangle$

 $145 \quad \mbox{which results in a precessional forcing } F(t) \mbox{ with amplitude one and zero mean.}$

We implicitly account for a slow terrestrial organic carbon reservoir (soil) as "buried organic carbon". It is reasonable to assume that monsoon, or enhanced precipitation will favor primary production and soil formation. But this recent soil together with older soils and with petrogenic organic carbon (Galy et al., 2008) will be eroded and transported to the ocean through enhanced river discharges. If the corresponding carbon is remineralized in the ocean without too much burial in the

150 alluvial fan, the net perturbation of the burial flux is likely to be negative (ie. net "old" soil erosion and remineralization). We will refer to this case as the "Amazon-like" situation, with the perturbation F(t) being subtracted to the baseline burial  $B_0$  by writing  $B = B_0 - a F(t)$ . In contrast, if most of the organic carbon is buried and preserved in the sediment, then the perturbation is likely to be positive, since it induces a net "recent" soil formation and burial. We call this the "Himalayan-like" situation, with now  $B = B_0 + a F(t)$ . Before 1 MyrBP and the associated major sea level changes, the river fans and

155 continental shelves should evolve mostly in a progradational way (see scheme on Fig. 3), a situation which a priori favors

5

Didier Paillard 8/8/y 13:29 Commentaire [6]: Instead of "fractionation". Cf. Peter Köhler's comment #1

Didier Paillard 8/8/y 11:58 Commentaire [7]: Rev.#2, RC2

& Peter Köhler's comment #1 Didier Paillard 8/8/y 11:59

**Commentaire [8]:** Peter Köhler's comment #3

Didier Paillard 8/8/y 11:58 Commentaire [9]: Rev.#2, RC4 & Peter Köhler's comment #2

Didier Paillard 9/8/y 17:19 Commentaire [10]: Peter Köhler's comment #5

Didier Paillard 9/8/y 17:22 Commentaire [11]: Peter Köhler's comment #4

Didier Paillard 8/8/y 12:08 Commentaire [12]: Rev.#1 organic carbon remineralization, while aggradational situations are likely to be more frequent in the late Pleistocene, with therefore a possible temporary reversal of the organic carbon burial.

**3** Results**

Our first simulations, with  $B = B_0 - a F(t)$ , correspond to a perpetual "Amazon-like" situation. They correspond to

- 160 experiment *a* (black lines) with no trend in the total carbon, and experiment *b* (blue lines), with an explicit linear trend in carbon. [The value of the parameter *a* is chosen in order to obtain approximately the correct amplitude for these simulated 400-kyr oscillations (a = 50 TgC/yr). Still, as can be seen on Fig.2, we obtain a surprisingly good match between the simulated and observed  $\delta^{13}$ C, with overall very similar cycles. More specifically, the  $\delta^{13}$ C black and blue simulated curves are superimposed and almost undistinguishable, since the linear trend added to the carbon cycle has almost no impact on the
- 165  $\delta^{13}$ C. They are both most of the time within the range of observed values (gray curves). The two main exceptions occur at about 0.3 and 2.3 MyrBP, with the simulated  $\delta^{13}$ C being significantly too high. In experiment *a* (black lines), pCO2 is oscillating around its equilibrium value, with two significant negative excursions occurring near 2.5 MyrBP and near 0.5 MyrBP. These lower values are directly linked to the ~2.4 Myr modulation of eccentricity. Obviously, with fixed or periodic parameters, this model cannot simulate a long term decreasing trend in carbon. When explicitly adding such a linear
- 170 decreasing trend (experiment b, blue lines), the two minima described above become two decreasing steps. The first one, occurring around 2.8 MyrBP, is coincident with the Plio-Pleistocene transition and the development of Northern hemisphere glaciations. The second one near 0.8 MyrBP is coincident with the Mid-Pleistocene transition (MPT) and the significant amplification of glaciations. Note that the timing of these two steps is directly linked to the astronomical forcing: it does not depend at all on the specifics of the trend that we used here. Two similar pCO2 decreasing episodes are also seen in the data
- 175 (Figure 1) though it is difficult to associate them with a precise timing, due to the difficulties to reconstruct accurately  $pCO_2$  from indirect proxies.

In order to account for the observed departure of the  $\delta^{13}$ C oscillations from a simple eccentricity forcing, we need to introduce a retroaction of Quaternary sea level changes onto the sedimentary dynamics of alluvial fans and continental shelves, and consequently onto organic carbon burial. As explained above, we will reverse the sign of our burial flux perturbation, and change it into  $B = B_0 + a F(t)$  when some conditions are met on the geomorphology of river outputs. In

- 180 perturbation, and change it into  $B = B_0 + a$  F(t) when some conditions are met on the geomorphology of river outputs. In particular it is necessary to account for a changing reservoir size that can be filled with sediments in an aggradational way. Indeed, at the first major sea level drop, rivers are incising though the river and fan bedrock, thus providing room for the accumulation of sediments loaded with organic carbon. This volume should be filled progressively with sedimentary organic carbon up to a point when further river incision, and consequent aggradation of sediment, do not affect the global organic
- 185 carbon anymore but only move sedimentary carbon from one place to the other. In other words, we will assume that the global "Himalayan-like" situation (ie. net organic carbon burial) is only a transient situation, linked to the first occurrence of a sea level minima. In order to illustrate this mechanism, we add a new equation for the slow geomorphological reservoir S

6

Didier Paillard 8/8/y 12:13 Commentaire [13]: Rev.#2, RC5

Didier Paillard 8/8/y 12:20 Commentaire [14]: Rev.#2, RC13 for organic carbon in river beds or river fans. We define its maximal size  $S_{MAX}$  from the observed sea level changes using the reference stack LR04 (Lisiecki and Raymo, 2005) by finding the previous sea level minima  $z_{MIN}$  (ie. the lower envelope) with the scaling  $S_{MAX} \sim z_{MIN}^3$  since it represents a volume of sediment (see Fig. 2).

In other words, the sedimentary organic carbon reservoir S grows at the pace of the above mentionned astronomical perturbation F0(t) up to its maximal size SMAX. After a short transient period, this reservoir remains therefore equal to this maximum value SMAX in the absence of major sea level drops, as during the pre-Quaternary period. In contrast, for each significant sea level drop, SMAX increases abruptly and we start a new transient phase whose duration is linked to parameter *b*. When S is small compared to the maximal reservoir size SMAX, then the aggradiational scheme is favoured, with river beds and deltaïc net organic carbon burial. But when S is close to its maximum value, we switch back to a mostly progradational sedimentation scheme, meaning that potential sea level changes will no more affect net global organic carbon burial :

200

190

(3b)

 $\mathbf{B} = \mathbf{B}_0 - a \mathbf{F}(\mathbf{t})$

 $B = B_0 + a F(t)$

Using this simple crude criteria, we obtain the results show on Fig. 2 (experiment *c*, red lines). As expected, this simple model does switch from the background "Amazon-like" or progradational burial mode to a "Himalayan-like" or aggradational mode, after each significant sea level drop, and most notably at two time periods, the first one between 2.4 and

- 205 2.5 MyrBP (as a consequence of the Plio-Pleistocene transition) and the second and largest one between 350 and 650 kyrBP (as a consequence of the MPT). The start of these transient periods is directly linked to sea level drops, according to the LR04 forcing, while the duration of these transients is linked both to the 0.85 SMAX threshold and the *b* parameter, whose values are chosen to qualitatively better match the  $\delta^{13}$ C data. For the results show on Fig.2, *b* = (160 kyr)4. Indeed, this sedimentary switch mechanism allows for a much better agreement with measured  $\delta^{13}$ C around 0.3 and 2.3 MyrBP, while
- 210 the first simulations were systematically too high at this time, as illustrated by the difference between the blue and red curves on Fig.2. We also simulate correctly the  $\delta^{13}$ C maximum around 500 kyrBP and the occurence of two broad "500 kyr" cycles over the last million years. With this burial mode switching mechanism, we are also able to predict an absolute minimum in carbon content, or long-term pCO2, around 600 kyrBP, in rather good agreement with the long term trend of pCO2 measured in Antarctic ice cores. Indeed, pCO2 from the Dome C record is about 5 to 10 ppm lower before the MPT (between 400 and
- 215 800 kyrBP), which is also what we obtain in our experiment *c*.

if  $S < 0.85 S_{MAX}$ :

otherwise:

**3 Discussion**

When variations in B, as determined by parameter *a*, are smaller than the baseline value B0, the model cannot reproduce the oceanic amplitude of  $\delta^{13}$ C observed in marine benthic records. The observed 400-kyr signal in  $\delta^{13}$ C records therefore requires major changes in the organic carbon burial, with almost no global net burial, but net oxidation episodes, during

7

**Didier Paillard 8/8/y 13:34 Commentaire [15]: Rev.#2, RC6**

Didier Paillard 8/8/y 15:42 **Commentaire [16]:** Rev.#2, RC6 Didier Paillard 9/8/y 13:15 **Commentaire [17]:** Rev.#2, RC7 Didier Paillard 8/8/y 15:52

Commentaire [18]: Rev.#2, RC8 Didier Paillard 9/8/y 13:18

qualitative match between measured and simulated  $\delta^{13}C$ , we can also add artificially a component linked to glacial-interglacial cycles. This is done by adding the detrended sea level LR04 curve to  $\delta^{13}C$  obtained from experiment c (orange line on figure 2). This allows to account for the significant 100-kyr periodicity seen in the carbon isotopic record, and usually attributed to glacial-interglacial changes in the global biospheric size (eg. Shackleton, 1977). - Didier Paillard 8/8/y 15:40

Didier Paillard 8/8/y 12:15 Commentaire [19]: Rev.#2, RC5 maxima of precessional forcing. This strong forcing therefore implies significant oscillations in the Earth carbon cycle for this time frequency, up to 4 or 5% in total carbon content. This is translated here into 10 to 20 ppm variations of  $pCO_2$  using our simple scaling, but it is very likely that these changes would be much larger, when accounting for interactions between  $pCO_2$  and climate. Indeed, colder climates are more favorable to oceanic carbon storage, as observed during the last glacial

240 cycles. According to this mechanism, in the ordinary sedimentary regime (progradation), we obtain changes in the carbon cycle with pCO2 maxima and  $\delta^{13}$ C minima associated directly to eccentricity maxima. This is indeed consistent with long Cenozoïc records (eg. Pälike et al, 2006).

When we allow for changes in the sedimentary regime triggered by sea level changes, the model can also reproduce more peculiar features. Indeed, up to now it has been difficult to explain the last two long-term cycles observed in the marine  $\delta^{13}$ C,

245 each being approximately 500 kyr-long, with a maximum now (δ13Cmax-I), a well-marked maximum at about 500 kyr BP (δ13Cmax-II) and a previous one around 1000 or 1100 kyr BP (δ13Cmax-III). In the model described here, these two long oscillations are generated from the eccentricity forcing, but with an abrupt switch to aggradation mode at about 620 kyrBP caused by the sea level drop at MIS 16. This switch reverses the phase of the 400-kyr carbon oscillation during a few hundred thousands of years. Interestingly, this also induces a slight minimum in the carbon (or pCO2) results, consistent with the observed low pCO2 values observed in the Antarctic ice core around 600-700 kyrBP. [This mechanism also allows for simulated marine δ13C in better agreement with data at about 2.4 MyrBP.]

It was already noted (Wang et al., 2004) that the climatic evolution since the last million years, in particular the Mid-Pleistocene Transition (MPT, about 0.8 MyrBP) and the Mid-Brunhes Event (MBE, about 0.4 MyrBP) were associated with the carbon isotopic maxima  $\delta^{13}$ Cmax-II and  $\delta^{13}$ Cmax-III. This was a strong indication of a possible causal link between the

- 255 long-term well-recognized eccentricity forcing on the carbon cycle and the Plio-Pleistocene climatic evolution. There is therefore a strong incentive to build a mechanistic astronomical theory of the carbon cycle. But a prerequisite towards understanding this long-term precessionally forced carbon cycle and its climatic consequences is to explain the observed changes during the Quaternary, in terms of  $\delta^{13}$ C and simultaneously in the atmospheric CO2 levels (Lüthi et al, 2008). The model results outlined above are a first step in this direction.
- 260 As detailed above, the fact that the 400-kyr carbon isotope cycle is perturbed during the Pleistocene strongly points towards a major role for organic matter burial over continental shelf areas being affected by sea-level changes. Obviously, this model is far too simple to represent faithfully the complexities of sedimentary dynamics in coastal areas, its consequences on organic matter preservation, on carbon cycle and ultimately on climate. Besides, we provide here no explanation for the prescribed multi-million year decreasing trend in pCO2. There is unfortunately no clear consensus on the actual mechanisms
- 265 involved, though this trend has been often attributed to long-term changes in continental weathering linked either to mountain uplift (Raymo and Ruddiman, 1992), to continental drift or mantle degassing rate (Lefebvre et al., 2013). Furthermore, we considered only sea-level changes as a potential feedback on organic matter burial in coastal areas. Obviously, many other important climatic feedbacks would also play a role: for instance increased temperatures would

8

Didier Paillard 9/8/y 14:32 Commentaire [20]: Rev.#2, RC10 Didier Paillard 9/8/y 13:28 Commentaire [21]: Rev.#2, RC8 probably reduce net primary production as a consequence of increased stratification, and therefore reduce organic carbon

270 deposition in coastal sediments, but it would also decrease oxygen concentrations and consequently would favor organic matter preservation. Similarly, stronger monsoons events would enhance the delivery of nutrients to the continental shelves, and therefore biological productivity. This would in addition deliver more fine-grained clay minerals that are necessary to seal and preserve organic matter from oxidation. This would work opposite to our continental soil-carbon mechanism for which enhanced monsoons lead to more organic carbon oxidation in agreement with the isotopic records. But, as a proof of concept, our model is chosen as minimalistic as possible. It does not attempt to include all potentially important mechanisms.

**275 concept, our model is chosen as minimalistic as possible. It does not attempt to include an potentially important mechan**

**4 Conclusion**

Our basic assumptions are primarily based on recent re-assessments of riverine organic carbon inputs to the ocean. With the above conceptual model, we demonstrate that simple mechanistic assumptions can account for the major patterns of the observed global evolution of carbon and carbon isotopes over this time period: First, enhanced precessional forcing linked to

- 280 high eccentricity leads to more continental organic carbon been washed out and remineralized, therefore a net decrease in overall organic carbon burial. Second, this mechanism is temporarily reversed following major sea-level drops associated with glaciations. [This model was built on the premises that changes in organic matter or petrogenic organic carbon fluxes are responsible for the 400-kyr oscillations observed in Cenozoïc 13C records, and that the large sea-level variations occurring during the Quaternary are strongly affecting this process. Continental margins and sedimentary fans are a very likely key
- 285 component, as illustrated by our simple conceptual model. But obviously, many complex processes are involved in the interactions between organic matter burial or oxidation, monsoons and sea-level changes. The geomorphological mechanism described here is one possibility which allows, for the first time, to account both for the persistent 400-kyr oscillation observed in 13C records during the Cenozoïc, but also for its change during the last million years. It also suggests the occurrence of possibly significant CO2 drops at about 0.8 MyrBP (Mid-Pleistocene transition) and at about 2.8 MyrBP (Plio-
- 290 Pleistocene transition), that would ultimately link the timing of these transitions to the astronomical forcing. Our model also provides a possible explanation for the puzzling shifted level in the CO2 records associated with the MBE

[revised manuscript text omitted]